cognition, behaviour

social norms, reinforcement learning, social cognition

**Author for correspondence:**
Uri Hertz
e-mail: uhertz@cog.haifa.ac.il

# Learning how to behave: cognitive learning processes account for asymmetries in adaptation to social norms

Uri Hertz

Department of Cognitive Sciences, University of Haifa, Haifa 3498838, Israel

UH, 0000-0003-4852-3516

Changes to social settings caused by migration, cultural change or pandemics force us to adapt to new social norms. Social norms provide groups of individuals with behavioural prescriptions and therefore can be inferred by observing their behaviour. This work aims to examine how cognitive learning processes affect adaptation and learning of new social norms. Using a multiplayer game, I found that participants initially complied with various social norms exhibited by the behaviour of bot-players. After gaining experience with one norm, adaptation to a new norm was observed in all cases but one, where an active-harm norm was resistant to adaptation. Using computational learning models, I found that active behaviours were learned faster than omissions, and harmful behaviours were more readily attributed to all group members than beneficial behaviours. These results provide a cognitive foundation for learning and adaptation to descriptive norms and can inform future investigations of group-level learning and cross-cultural adaptation.

## 1. Introduction

Social norms are the unwritten rules that prescribe and guide behaviour within a society and with which group members generally comply [1–3]. Social norms govern a group's behaviour, are manifested in the behaviour of most individuals most of the time and may change between social groups and over time. For example, the norm governing how we greet each other when we meet can differ quite arbitrarily from one culture to the next, or during global events such as the COVID-19 pandemic (figure 1). Adhering to group norms can ensure cooperation within a group [2,4], make social conduct more predictable [5] and signal one's group affiliation to others [6]. Failure to learn and adapt might unintentionally send the wrong signals through inappropriate behaviour that may lead to frustration, isolation, resentment and intergroup distress [7]. While the challenge of learning and adapting to new social norms has been studied from the perspective of the social structures and mechanisms supporting socialization [8] as well as from an evolutionary, normative point of view [1,2], far less attention has been devoted to the contribution of social cognitive learning mechanisms to this problem.

Social norms change how individuals behave. Such norms include injunctive norms, which indicate how people *should* behave, and descriptive norms indicate how other people behave [9], which are at the focus of this work. Descriptive norms have been shown to affect people's behaviour, for example, when exposed to other people's recycling habits [10], finance management [11] or alcohol use [12]. Such norm effects have also been observed in laboratory experiments, notably in the seminal works on social influence and conformity by Sherif [13] and Asch [14] regarding perceptual decisions. Other studies have shown that people adapt their behaviour and preferences after learning about others' preferences [15–17], indicating the importance of social information in

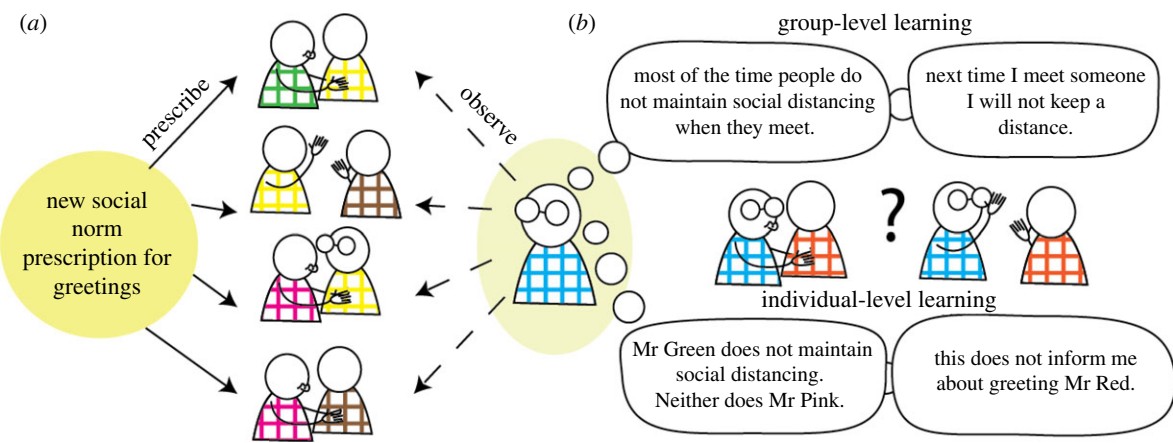

**Figure 1.** Learning a new social norm's behavioural prescription. When introduced to a new social setting, one may need to adapt one's behaviour according to the prevalent social norm. (*a*) Such social norms stochastically govern the behaviour of individuals in a group, affecting most group members most of the time. Newcomers (e.g. the person in blue plaid in the figure) can infer the social norm through accumulated observations and experiencing of interactions between group members and can adapt their behaviour accordingly. (*b*) Such learning can occur at a group level (top), i.e. attributing the behaviour to all group members, or on an individual level (bottom), i.e. learning only about specific individuals. (Online version in colour.)

forming one's own behaviour and beliefs [18–20]. While it is possible to explicitly state a descriptive norm, in many cases, people form their perception of norms on their own [21]. The effects of social norms on behaviour may therefore rely on how people learn about others' behaviour and form such a descriptive norm.

One way to learn about a descriptive social norm is by observing the behaviour of members of a group, and accumulating such observations over time [22–26] (figure 1a). Such accounts borrow from non-social computational models of associative and reinforcement learning [22]. For example, when learning about a person's honesty, one may observe whether a person gives truthful advice over time, increasing the estimation of her honesty when she gives accurate advice, and decreasing it when she gives misleading advice [27]. When learning about groups, learners may use the same learning mechanisms, learning about specific individuals in a group, and adjust their behaviour according to the specific partner they encounter. However, learners may learn a group-level trait, attributing observations from individuals to all group members, indicating learning about a social norm that governs the group's behaviour [28–31] (figure 1b).

In the literature concerning learning about action-outcome associations, such as Pavlovian and operant conditioning, the strength of associative learning is often mapped to two dimensions—the appetitive/aversive outcome of an action and the active/passive nature of the action [32,33]. For example, one may learn to increase a pattern of behaviour after it has been actively rewarded, or when it leads to the omission of an aversive response (avoiding punishment). Similarly, the omission of an appetitive outcome and receiving punishment may lead a learner to reduce the likelihood of displaying a behaviour pattern. While these contingencies may rely on similar computational principles, they are known to be processed differently. Punishments and rewards are processed by different neural mechanisms [34] and can have different effects on learning. Similarly, omission and action are perceived and processed differently [35,36]. Such asymmetries can therefore give rise to different biases in learning and shape the way people learn and adapt to social norms.

This work seeks to examine how features of the behavioural prescription of social norms affect adaptation to these norms and the learning process constraints underlying such effects. Specifically, it is hypothesized that due to constraints of the cognitive learning mechanisms, behavioural features of social norms will make some norms easier to attain and harder to relinquish in favour of new norms. One constraint has to do with the perceptual aspects of learning, as some behaviours are more readily detected than others, e.g. action versus omission. In addition, the transfer from individual-level learning to group-level learning may be influenced by the norm's behavioural prescriptions. For example, as negative moral behaviour is more readily attributed to an individual's character than positive behaviour [37], behaviours with aversive outcomes may be more readily generalized to all group members than helping behaviours.

To study these hypotheses, I adapted the appetitive/ aversive and action/omission dimensions to the domain of social norms, using norms that prescribe behaviour that can benefit/harm others through action/omission acts (figure 2). In a sequential social dilemma paradigm called the Star-harvest Game [38], participants collected stars and could sacrifice a move to zap other players. In different experimental conditions, zap outcomes were either harmful or beneficial to other players. The participants were exposed to different social norms displayed by the behaviour of three bot-players. The action/omission dimension was formed by the bot-players' active zapping or zap avoidance behaviour (figure 2). Different combinations of these features formed different types of norms, which were characterized by different behavioural prescriptions. The Harm-Action norm was marked by active zaps that had negative outcomes for others, i.e. zapping a player who is on your route to a star. The Harm-Omission norm was manifested in avoidance of negative zaps. The Benefit-Action norm was manifested in active zaps that had positive outcomes for others, while the avoidance of positive zaps was a manifestation of the Benefit-Omission norm. This allowed examination of how participants learn and adapt to social norms and which social norms persist when moving to a new social environment.

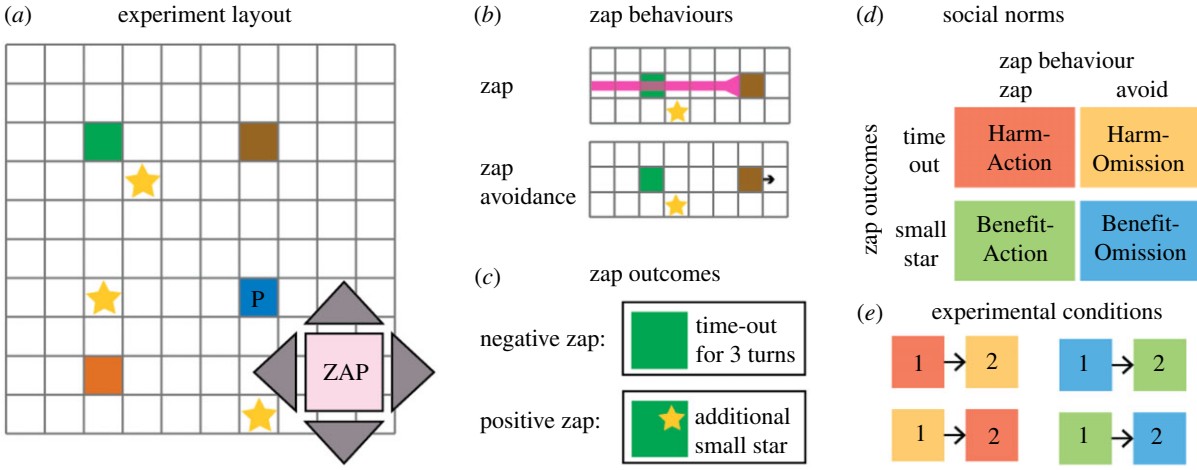

**Figure 2.** Experimental design—the Star-harvest Game. Participants played the Star-harvest Game online. (*a*) The game layout consisted of four players who moved across a two-dimensional grid and collected stars, with players marked by coloured squares. The participant in this case played the blue square (marked P), playing against three other bot-players. In each trial, players could either move using the blue arrows or zap using the arrows and the zap button. (*b*) Players could either zap each other by sending a ray that affected other players or avoid zapping other players. (*c*) The zap outcome was either harmful, sending the zapped player to a time-out zone for three turns, or beneficial, such that the affected player received a small star worth a tenth of a regular star. (*d*) The algorithms governing the behaviour of the bot-players led to four different social norms with different behavioural prescriptions. (*e*) Participants played two experimental blocks, with different bot-players displaying different norms. The zap outcome remained consistent over the two blocks. (Online version in colour.)

## 2. Methods

### (a) Participants

The Amazon M-Turk platform was used to recruit 276 participants for this study, including 157 men (age: mean ± s.d.: 35.45 ± 9.75) and 119 women (age: 39.12 ± 11.1). Participants were randomly assigned to one of four experimental conditions, which differed in the order of experimental blocks and the zap outcome (positive/negative). I aimed for at least 60 participants in each of the four conditions based on estimation of an effect size of 0.5 for a within-participants difference in zapping rates between Harm-Omission and Harm-Action conditions, based on a pilot of 20 participants (not included in this study). Due to the random assignment and to ensure there were enough participants in each condition, the number of participants in each block differed slightly (Harm-Action First $N = 76$, Harm-Omission First: $N = 71$, Benefit-Omission First: $N = 63$, Benefit-Action First: $N = 66$). No participant was excluded from the analyses. All participants provided informed consent and received monetary compensation at a fixed rate of 3.5 USD for 15 min of participation. The study was approved by the research ethics committee at the Faculty of Social Sciences at the University of Haifa, Israel (number 038/18).

### (b) Star-harvest Game

The Star-harvest Game was developed to provide a flexible and rich setting in which multiple types of social norms can be displayed in a user-friendly manner. The game included four players, represented by coloured squares that move around a $10 \times 10$ grid (figure 2, see example here: http://socialdecisionlab. net/resources.html). The game is played on a turn-by-turn basis, and the order of players remains constant throughout the game. In each turn, the players could either move in one of four directions and collect stars that appeared on the grid, or zap by emitting a pink ray in one of the four directions. Players caught in the ray in the negative zap outcome conditions were sent to a 'time-out zone' visible to the player for three turns. Those caught in the ray in the positive zap outcome conditions received a small bonus star. After each round in which all players took a turn, a new star could appear somewhere on the grid with a 0.75 probability, and uncollected stars could disappear. Each player's collected stars appeared in their 'score' section on the screen. The participants did not receive any bonus based on the stars they collected beyond the fixed monetary rate.

### (c) Social norms algorithms

The behaviour of the bot-players was governed by algorithms implementing different social norms. In each experimental condition, the behaviour of all three bot-players was governed by the same algorithm. A short description of the different algorithms is given below, and a detailed description of the algorithms is provided in the electronic supplementary material.

All bot-players began each turn by looking for stars. If they were the player closest to a star, they would move towards it. Otherwise, when the zap outcome was negative, Harm-Action bot-players zapped other players that were on their way to a star, while Harm-Omission bot-players would move away from other players without zapping. When the zap outcome was positive, Benefit-Omission bot-players would also move away without zapping anyone. Benefit-Action bot-players would start every turn with a probability of zapping others, even if they were closest to a star. This probability was dependent on their distance from the closest star and decreased the closer they were to the star (distance of 1 was associated with a zap probability of 0.02)

### (d) Analysis

Statistical analyses were carried using Matlab R2018b (Mathworks Inc., USA). The Markov chain Monte Carlo (MCMC) Metropolis–Hastings algorithm was used for model fitting and estimation for each participant [39]. For model comparisons, for each model, a deviance information criterion [40] was calculated for each individual. I used in-house Matlab code and an MCMC toolbox for Matlab developed by Marko Laine [41].

## 3. Results

Participants played the Star-harvest Game online, where they moved across a two-dimensional grid using arrows and collected stars that appeared (and disappeared) from time to time, with three other bot-players (figure 2). Participants

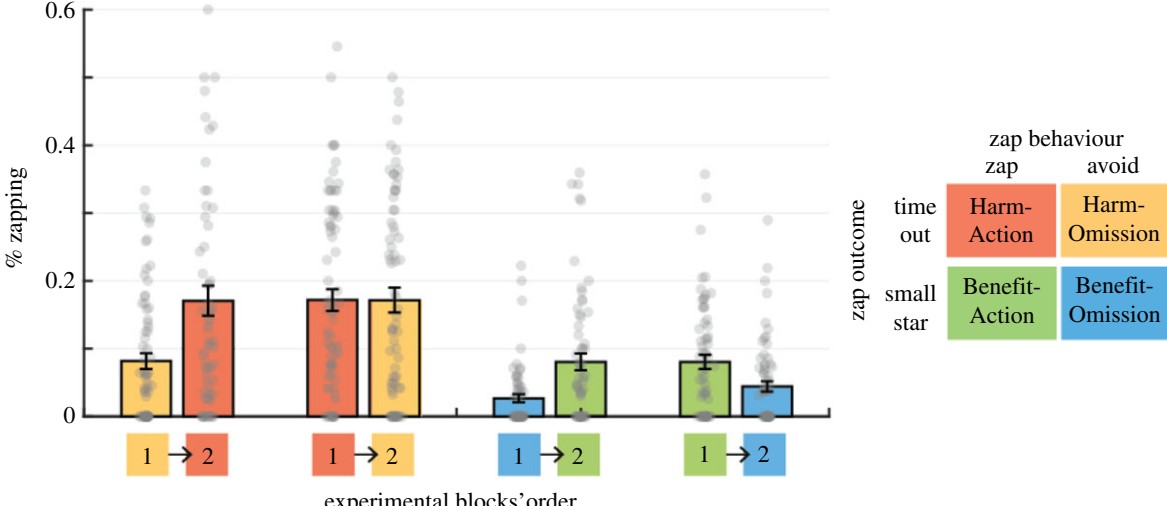

**Figure 3.** Adaptation to new social norms. Participants played the Star-harvest Game under four experimental conditions, in two blocks that differed according to the social norms displayed by the bot-players. In each experimental condition, the percentage of times participants zapped others when they had the opportunity to do so was examined. In the first experimental block, participants adapted their zapping behaviour and matched the zapping norm around them. In the second block, participants adapted their behaviour to the new norm during all transitions except the transition from Harm-Action norm to Harm-Omission norm (red to yellow bars). On all the graphs, grey dots represent individual scores, bars represent the mean, and error bars represent the standard error of the mean. (Online version in colour.)

were randomly assigned to one of the four experimental conditions. Each experimental condition began with one experimental block in which the three bot-players displayed one of the four norms (Harm-Action, Harm-Omission, Benefit-Action, Benefit-Omission), followed by a second block with a new set of bot-players, marked by changes in the players' colours, which displayed a different norm. Each experimental block included 70 turns for each player. To make the experimental instructions consistent, the zap's outcome did not change between blocks, such that the norms displayed by the bot-players changed from Harm-Action to Harm-Omission (and vice versa) or from Benefit-Action to Benefit-Omission (and vice versa).

The main behavioural marker of adaptation to social norms was the percentage of times participants zapped other players when they had the opportunity to do so, i.e. when they shared a column or row with another player (see numbers of zaps and zap opportunities in electronic supplementary material). Adaptation to the norm would result in lower zapping rates when the bot-players avoid zapping (Harm-Omission and Benefit-Omission norms) than when bot-players actively zap others (Harm-Action and Benefit-Action norms). In the first experimental block, the participants in all conditions adapted their zapping behaviour to the behaviour of their surroundings (figure 3). This adaptation was quantified using an ANOVA, with zapping percentages as the dependent variable and Zap Outcome (Harm/Benefit) and Zap Behaviour (Action/Omission) and their interactions as main effects. I found a significant effect of Zap Behaviour ($F_{1,272} = 35.92$, $p < 0.0001$, partial $\eta^2 = 0.115$), indicating that participants were more likely to zap others when they were in the company of other zappers. In addition, participants were more likely to zap others when zaps were associated with harmful outcomes ($F_{1,272} = 37.09$, $p < 0.0001$, partial $\eta^2 = 0.122$), indicating a bias towards competitive behaviour in such video-game scenarios. The interaction between Zap Behaviour and Zap Outcome was not significant ($F_{1,272} = 2.29$, $p = 0.13$, partial $\eta^2 = 0.008$).

These results indicate that in the first experimental block, lacking prior experience in the task, participants generally learned and adapted to all social norms.

The next behavioural analysis examined adaptation to new social norms between the first and the second experimental blocks by subtracting each participant's zapping rate in the omission norm block from the action norm block. When this measure was positive, it indicated high behavioural adaptation between conditions, in line with the change in norms. When it was close to zero, it indicated low behavioural adaptation. An ANOVA was used to analyse the individual adaptation patterns, with Zap-Behaviour Order (Action First/Omission First), Zap-Outcome (Benefit /Harm) and their interaction as main effects. A significant Zap-Behaviour Order effect was found ($F_{1,272} = 12.65$, $p = 0.0004$, partial $\eta^2 = 0.044$), indicating that participants displayed higher levels of adaptation when moving from an omission norm to an action norm. In addition, a significant interaction was found between Zap-Norm Order and Zap-Outcome ($F_{1,272} = 5.73$, $p = 0.017$, partial $\eta^2 = 0.02$), with no significant Zap-Outcome effect ($F_{1,272} = 0.00002$, $p = 0.97$, partial $\eta^2 < 0.0001$). As can be seen in the averaged zap-rate graph (figure 3), participants showed adaptations in all conditions but one—when moving from a Harm-Action norm to a Harm-Omission norm, giving rise to the interaction effect.

The next analysis steps were aimed at examining potential learning mechanisms that underlie the behavioural adaptation patterns, using computational learning models. The models were used to examine how zap behaviour (action/omission) and zap outcome (benefit/harm) are treated by a learner, in line with the asymmetries in adaptations observed so far. In addition, the models were designed to examine whether and how observation of one player's behaviour are used to infer group-level norms (figure 4). Specifically, the models included individual-level learning, group-level learning and a hybrid biased-attribution model that allows weighted level of attribution from individual to group level (electronic supplementary material; figure 4).

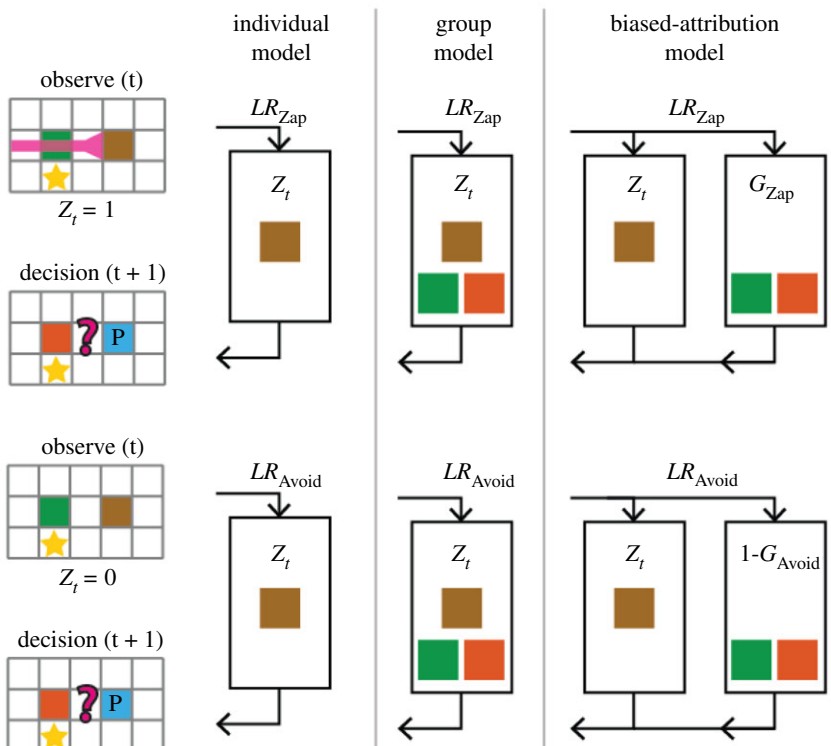

**Figure 4.** Models of learning about others' behaviour. After observing a player's behaviour, the participants may update their beliefs about other players' likelihood to zap, which would inform their future decisions whether to zap others. Three learning models were suggested. In the individual-model, learning is done on the individual level, and the behaviour of the observed player is used to update belief about his zap behaviour with value $Z_t$, with learning rate $LR_{Zap}/LR_{Avoid}$ depending on the observed behaviour (zap/avoidance). In the group-model, one player's behaviour is used to update all players' zap probability with the same values and learning rates. In the hybrid biased-attribution model, the observed player's and the other players' zap behaviour are both updated with the same learning rates but with different values. (Online version in colour.)

All models were aimed at predicting the participants' decision to zap a target player, i.e. a player that shares a column or row with the participant. This decision on each trial $t$ was logistically dependent on a number of variables (equation (3.1)): the participant's overall tendency to zap, his current distance from a star (variable *StarDist*), his current distance from the target player (variable *TargetDist*), and the estimated zapping behaviour of this target player (the probability that the target would zap other players, parameter *ZapProb*). The contribution of these variables to the decisions was determined by a set of free parameters $\{w_0, w_1, w_2, w_3\}$, and the value of these variables was calculated in each turn. These weights were used to model the cost associated with zapping, as they allow the availability of stars to overcome the tendency to zap others.

$$p_t(\text{Zap a target}) \sim w_0 + w_1 \cdot \text{StarDist}_t + w_2 \cdot \text{TargetDist}_t + w_3 \cdot \text{ZapProb}_t^{\text{Target}}). \quad (3.1)$$

The distance to stars and targets can be calculated directly from the data available in each turn. However, the target player's zapping behaviour, i.e. the probability that the target player would zap other players, had to be learned from observations and interactions in previous turns. This learning mechanism differed between models (figure 4). In all models, no learning was done if the observed player did not have an opportunity to zap anyone, i.e. did not share a row or a column with any player, or if the observed player was the closest player to a star and moved toward it. In addition, the models were fitted to the data with no information regarding the

outcome of the zaps, harmful or beneficial, and were affected only by the estimated likelihood of zapping and distance to stars and targets.

The first model assumed that learning occurred only at the individual level. When observing player $p$'s zap (or avoidance), the learner updates his belief about the likelihood of player $p$ to zap in the future (figure 4). When player $p$ zaps another player at time $t$, the variable $Z_t$ is set to 1, and when player $p$ avoids zapping (he had the opportunity but did not zap), $Z_t$ is set to 0. A prediction error is calculated between $Z_t$ and the previous estimation of the player's zapping probability, $\text{ZapProb}_t^p$, and is used to update this probability with different learning rates for zap and avoidance. Zapping probabilities for all players were initially set by another free parameter prior. To account for asymmetry in learning, the model included different learning rates for action (zaps) and omission (avoidance).

$$\text{ZapProb}_{t+1}^p = \text{ZapProb}_t^p + \begin{cases} LR_{\text{Zap}} \cdot (1 - \text{ZapProb}_t^p) & Z_t = 1 \\ LR_{\text{Avoid}} \cdot (0 - \text{ZapProb}_t^p) & Z_t = 0 \end{cases}. \quad (3.2)$$

The second model assumed a complete attribution to group level, where each observation is used to update a group-level zap probability which applies to all players, $\text{apProb}_t^{\text{Group}}$. Such transfer can speed up learning and adaptation to new norms, as it accumulates information across all players, and is especially useful when displays of the new

**Table 1.** Parameter estimations of the hybrid learning model separately for the negative and positive outcome conditions.

| | $w_0$ | $LR_{Zap}$ | $LR_{Avoid}$ | $w_1$ | $w_2$ | $w_3$ | $G_{Zap}$ | $G_{Avoid}$ | Prior |
|---|---|---|---|---|---|---|---|---|---|
| negative zap conditions ($N = 136$) | | | | | | | | | |
| estimate mean ± s.e.m. | −4.5 ± 0.078 | 0.41 ± 0.012 | 0.12 ± 0.01 | 1.53 ± 0.15 | −3.36 ± 0.21 | 4.59 ± 0.22 | 0.62 ± 0.015 | 0.57 ± 0.014 | 0.45 ± 0.014 |
| $t$(d.f. = 135) | −31.8 | | | 5.42 | −8.98 | 11.39 | | | |
| $p$ | <0.0001 | | | <0.0001 | <0.0001 | <0.0001 | | | |
| positive zap conditions ($N = 113$) | | | | | | | | | |
| estimate mean ± s.e.m. | −5 ± 0.11 | 0.44 ± 0.013 | 0.3 ± 0.015 | 1.3 ± 0.21 | −1.8 ± 0.23 | 3.74 ± 0.27 | 0.49 ± 0.014 | 0.71 ± 0.013 | 0.49 ± 0.015 |
| $t$(d.f. = 112) | −23.36 | | | 3.23 | −3.87 | 6.88 | | | |
| $p$ | <0.0001 | | | 0.0016 | 0.0002 | <0.0001 | | | |

norm's behaviour are sparse [29].

$$\text{ZapProb}_{t+1}^{\text{Group}} = \text{ZapProb}_t^{\text{Group}}$$
$$+ \begin{cases} LR_{\text{Zap}} \cdot (1 - \text{ZapProb}_t^{\text{Group}}) & Z_t = 1 \\ LR_{\text{Avoid}} \cdot (0 - \text{ZapProb}_t^{\text{Group}}) & Z_t = 0 \end{cases}.$$
(3.3)

The third model was a hybrid biased-attribution model that included the individual learning mechanism (equation (3.2)) and an additional group-level updating to all other players. This is captured by two free parameters $\{G_{\text{Zap}}, G_{\text{Avoid}}\}$, which specify how much each observed zap or avoidance behaviour of one player is attributed to the group, i.e. to the updating of the zap probability of the other players. When $G_{\text{Zap}}$ is set to 1, it increases the zap probability of other players as if these players were doing the zapping, converging with the group-model. When $G_{\text{Avoid}}$ is set to 1, it decreases the zap probability of other players as if they avoided zapping, again converging with the norm learning model. However, when $\{G_{\text{Zap}}, G_{\text{Avoid}}\}$ are close to 0.5 the transfer is non-informative, essentially setting an expectation that other players are just as likely to zap or avoid. Asymmetries in the generalization parameters would lead to biased attribution to group level, as is demonstrated in a set of simulations in the electronic supplementary material.

$$\text{ZapProb}_{t+1}^{\text{Others}} = \text{ZapProb}_t^{\text{Others}}$$
$$+ \begin{cases} LR_{\text{Zap}} \cdot (G_{\text{Zap}} - \text{ZapProb}_t^{\text{Others}}) & Z_t = 1 \\ LR_{\text{Avoid}} \cdot ((1 - G_{\text{Avoid}}) - \text{ZapProb}_t^{\text{Others}}) & Z_t = 0 \end{cases}.$$
(3.4)

All models were fitted to each participant's decisions (zap/avoid) across both experimental blocks (see methods and electronic supplementary material). To avoid unreliable parameter estimation, only participants who zapped at least once were included in this analysis ($N = 248$ of 276) (the mixed-effect analysis of adaptation in zap behaviour was carried on this subset of participants with no changes in the results from the main analysis, see electronic supplementary material). In a series of model comparisons, taking into account both model fit to the data and the number of parameters used by the model, the biased-attribution model was found to significantly outperform other models (see electronic supplementary material, table S1, including a model with different parameters for direct and indirect reciprocity

[42]). This result indicates that our participants did not use a reciprocity learning rule, but were flexible in the way they update beliefs about other players, i.e. group level or norm inference, from observation of single players.

The model fitting procedure allowed estimation of all free parameters for all participants, facilitating overall evaluation of these parameters and comparing them between groups of participants (table 1). The weights assigned to each factor affecting zapping ($w_0$, $w_1$, $w_2$, $w_3$) all significantly differed from 0 in both the positive and negative zap-outcome conditions (table 1). Overall, participants were averse to zaps (negative $w_0$), more so when zapping had a positive outcome than when it had a negative outcome ($p = 0.04$, table 1). Participants were more likely to zap when stars were far away (positive $w_1$), indicating the cost of zaps. They were more likely to zap targets that were close to them (negative $w_2$), more so when zaps had harmful outcomes than when they had beneficial outcomes ($p = 0.01$, table 1). These results indicate that the participants were sensitive to the task settings in each turn, their distance to stars and to other players, and these affected their decision to zap other players.

The biased-attribution model also indicated that participants were affected by other players' likelihood of zapping (positive $w_3$). This value was learned by observing the players' behaviour over time. The model included two learning rates, for action (zap) and for omission (zap avoidance) behaviours (figure 4a). The effects of Zap Behaviour (action/omission, within-subjects), Zap Outcome (harm/benefit, between subjects) and their interaction on learning rates, were examined using a mixed-effects ANOVA, with individually estimated learning rates ($LR_{\text{Zap}}, LR_{\text{Avoid}}$) as independent variables. A significant Zap Behaviour effect was observed ($F_{1,529} = 79.81$, $p < 0.0001$, partial $\eta^2 = 0.23$), such that learning rates for action (zaps) were higher than for omission (avoidance). In addition, a significant Zap Outcome effect was found ($F_{1,529} = 19.96$, $p < 0.0001$, partial $\eta^2 = 0.07$), such that participants in the beneficial zap conditions had higher overall learning rates. Finally, a significant interaction effect was observed ($F_{1,529} = 14.83$, $p = 0.0001$, partial $\eta^2 = 0.052$), such that learning from Harm-Omission behaviour was associated with lower learning rates than learning from Benefit-Omission behaviour.

In addition, two parameters were estimated for group-level attribution of information from the observed player to all other players (figure 5b). The effects of Zap Behaviour (action/omission, within subjects), Zap Outcome (harm/

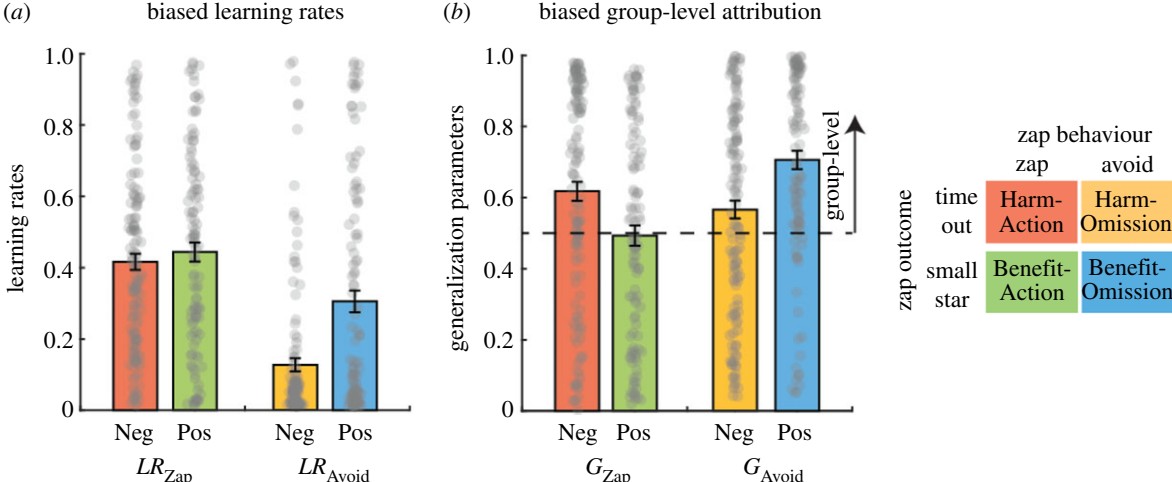

**Figure 5.** Learning parameters underlying asymmetric adaptation to social norms. The biased-attribution model that best fit participants' decisions to zap included two mechanisms for learning from observing other players' zaps or zap avoidance. (a) The model included different learning rates for action (zap) and omission (zap avoidance) behaviours. Learning rates were lower for omission than for action, indicating an asymmetry in the contribution of active behaviour to norm learning. (b) The model included different group-level attribution parameters for omission (zap avoidance) and action (zaps). Group-level attribution values were higher for behaviours that had an aversive contingency—Harm-Action and Benefit-Omission norms. In all the graphs, grey dots represent individual scores, bars represent the mean and error bars the standard error of the mean. (Online version in colour.)

benefit, between subjects) and their interaction on group-level attribution were examined using a mixed-effects ANOVA, with the individually estimated group-level transfer parameters as independent variables. A significant effect of Zap Behaviour was observed, such that omission (avoidance) behaviours were associated with higher group-level transfer values ($F_{1,529} = 10.53$, $p = 0.0013$, partial $\eta^2 = 0.038$). Zap Outcome did not have a significant effect ($F_{1,529} = 0.1$, $p = 0.75$, partial $\eta^2 < 0.001$). A significant interaction was observed ($F_{1,529} = 28.98$, $p < 0.0001$, partial $\eta^2 = 0.099$), pointing to higher group-level attribution of behaviours with aversive contingencies: Harm-Action and Benefit-Omission.

## 4. Discussion

The aim of this study was to investigate how cognitive learning mechanisms account for learning and adaption to new social norms. Specifically, I examined how two features of the behavioural prescription of norms, its manifestation in action or omission and the outcome of this behaviour, whether beneficial or harmful, affect learning and adaptation. Using a multiplayer Star-harvest Game in which the behaviour of three bot-players was governed by algorithms that implemented four different social norms, I examined how people learn new social norms and how their experience with one norm affects adaptation in the transition from one norm to another. I found that on their first encounter with the task, participants learned and adapted to the social norm displayed by the bot-players. Yet, in the second block of the experiment, when a new social norm was displayed by a new set of bot-players, their previous experience affected their adaptation. Specifically, the norm manifested in Harm-Action behaviour persisted when participants faced a new set of Harm-Omission players, while in all other transitions, participants adapted their behaviour. The resistant norm was characterized both by an active behaviour and by a harmful outcome for others, implying a competitive intent.

This combination seems to contribute to the unique persistence of this social norm in the current experimental design.

Computational modelling of social learning proposed a mechanistic explanation for the observed behaviour and indicated that social learning in the task went beyond individual-level learning. The best-fitted model, the biased-attribution model, suggested that participants' decisions to zap or avoid zapping other players were influenced by several parameters, among them the distance from stars, indicating the cost of zapping, and the estimation of the target player's likelihood to zap. This estimation was based on the specific player's previous behaviour, in line with the individual-level learning mechanism, and also incorporated other players' previous behaviour, in line with group-level inference. The weight given to other players' behaviour, or the magnitude of group-level attribution, was a free parameter in the model. Behaviours that carried an aversive contingency, omission of benefit or harmful action, were found to be more readily attributed to group level. Such group-level transfer facilitates learning as it allows rapid accumulation of sparse behaviours across players, instead of accumulating them independently for each participant. Behaviours with beneficial contingency were associated with low group-level attribution, suggesting more personal and reciprocity-based learning for prosocial behaviours [43]. In addition, learning rates associated with actions were consistently higher than for omissions, in line with non-social learning findings, further supporting the observed asymmetry in adaptation [36,44]. Both mechanisms can work together to make behavioural prescriptions persistent even when social settings change, attenuating adaptation to new social norms.

The results of this study are in line with previous findings on social learning of individuals' traits and behaviour, and demonstrate how these are linked to group-level inference findings. On the individual level, research has shown that people are quick to infer about bad social behaviour from sparse data, as such negative behaviours are deemed more diagnostic of a person's moral character [37,45]. In addition,

actions were shown to be more readily attributed and indicative of a person's general character than acts of omission, as they are both more likely to be detected and less likely to be explained away (plausible deniability) [36]. Social learning of individuals' traits was shown to be important to form predictions about others' behaviour and to adapt one's behaviour accordingly [24,46]. Beyond inferring from one's behaviour in a specific situation about his general trait, people also infer from one person to all other group members [30,47]. Adults and children can attribute a set of behaviours to all other group members, mostly when such group membership is salient [31,48]. The current results indicate that social learning about others' behaviour can be set on a continuum, with some behaviours more readily attributed than others on the individual level (action versus omission), and some more readily generalized to indicate group-level norm (harmful versus beneficial contingencies). A unified cognitive learning framework can account for both types of social learning, operating simultaneously for individual and group-level inference, and affecting adaptation and one's future behaviour.

The current study examines adaptation to norms in new, unfamiliar surroundings, the Star-harvest Game, and the effect of experience on adaptation to new norms. It, therefore, examines dynamic, quick, behavioural adaptation. It is a departure from studies aimed at characterizing cooperation and prosocial behaviour as a stable trait [49–51] or from examining gradual changes across development and acculturation [7,52]. The current study's approach is limited in the sense that the learned social norms may not represent a long-lasting behaviour or tendency, as it does not rely on real-life contexts, such as monetary or resource sharing, which are common in the study of social norms [28,53]. However, the current paradigm allows control of the effect of experience on social adaptation, and a rich and flexible laboratory model of social learning. As such, it may be useful for understanding cross-cultural differences in adaptation to social norms and the contribution of cognitive learning processes and cultural background (previous experience) to this process [53–55].

Some limitations arise from the use of bot-players instead of live-interaction with humans. Participants were not given explicit information regarding the identity of the other players, either if these were humans or bots. As the experiments were taking place online, where it is possible to interact anonymously both with other humans and with algorithmic bots, supported this ambiguity. While it is possible that during interactions with humans the patterns observed here will be amplified or different, participants'

behaviour in the positive zaps conditions, where zaps were mainly to benefit others and had no clear benefit for the participant, suggests that participants did treat the other players as if they were fellow participants, to some extent. Another limitation of the current experimental design was that the bot-players' behaviour was homogeneous, following the notion that social norms are carried by most group members, most of the times [2]. This meant that it was hard to distinguish between learning from first-hand experience (direct reciprocity) and second-hand observations (third-party reciprocity) [42], and examining how people learn in a non-homogeneous environment, with different people displaying different norms. However, future studies may build on the current paradigm and manipulate the rate of first and second-hand experiences, and the homogeneity of behaviour, as well as introducing groups and coalitions, to examine different social learning dynamics and their interaction with cognitive learning mechanisms [56,57].

To conclude, this study aimed to provide a cognitive learning perspective on the problem of learning and adaptation to social norms. The behavioural results indicated asymmetries in the learning of social norms, and the computational models indicated two mechanisms that may underlie these asymmetries. One such mechanism was an omission bias in learning, whereby actions were more readily learned than omissions. Another mechanism was a bias in group-level attribution, where behaviours with negative outcomes to others were more readily attributed to other group members than behaviours with positive outcomes. These mechanisms may influence adaptation to social norms outside the laboratory, making social norms whose behavioural manifestations are active and harmful more persist even when social settings change. Finally, the experimental approach used here can be elaborated to account for many different norms, and the use of principles and computational frameworks from cognitive learning can inform future investigations of cross-cultural differences and adaptation to descriptive social norms.

Ethics. The study was approved by the local research ethics committee (project 038/18) at the Faculty of Social Sciences at the University of Haifa, Israel.

Data accessibility. All datasets and code used to analyse the data are available at: https://osf.io/f6erm/. A demo of the Star-harvest Game is available at: http://socialdecisionlab.net/resources.html.

The data are provided in the electronic supplementary material [58].

Competing interests. We declare we have no competing interests.

Funding. U.H. was supported by the Israel Science Foundation (1532/20).

Acknowledgement. The author thanks Prof. Chirs D. Frith for his comments and feedback on an earlier version of this manuscript.

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
