## [Peer Review File · Proceedings of the Royal Society B: Biological Sciences]

Review History

RSPB-2021-0293.R0 (Original submission)

Review form: Reviewer 1

Recommendation

Accept with minor revision (please list in comments)

Scientific importance: Is the manuscript an original and important contribution to its field?

Acceptable

General interest: Is the paper of sufficient general interest?

Excellent

Quality of the paper: Is the overall quality of the paper suitable?

Excellent

Is the length of the paper justified?

Yes

Should the paper be seen by a specialist statistical reviewer?

No

Do you have any concerns about statistical analyses in this paper? If so, please specify them explicitly in your report.

No

It is a condition of publication that authors make their supporting data, code and materials available - either as supplementary material or hosted in an external repository. Please rate, if applicable, the supporting data on the following criteria.

Is it accessible?

Yes

Is it clear?

Yes

Is it adequate?

Yes

Do you have any ethical concerns with this paper?

No

Comments to the Author

This paper presents data collected in an online game alongside a computational model to demonstrate that influence from group-level behavior affects the behavior of participants above and beyond their interactions with other individual agents (e.g. in reciprocal exchanges of rewards/punishments). Players controlled and agent and navigated in a 2D space, collecting star rewards and delivering “zaps” that had either positive or negative effects on AI controlled agents. I really enjoyed this paper and believe the models and the method can contribute to our understanding of norms. I believe it would be a strong publication in *Proceedings of the Royal Academy B*, as long as a few points can be clarified and/or elaborated on.

1. Most significantly, it would help to have a stronger test of agent-based influence included in the author’s computational model. His model of reciprocity-based learning appears to model the influence of AI agent’s “zapping” on the participant’s “zapping” behavior by observing the AI’s use of “zaps”, but the distinction between second-party and third-party reward/punishment is not discussed. Would it be possible to separately model the influence of “zaps” that players personally receive from other agents (i.e. second-party reciprocity), while separately modeling the influence of “zaps” that players observe between the bots (i.e. third-party reciprocity)? It seems likely that players are more likely to reciprocate “zaps” to individual bots when they were directly affected by them, and modeling this make improve the fit of the reciprocity-based model.
2. The author describes his study as examining descriptive norms, which I agree is correct. However, descriptive norms are usually described in contrast to injunctive norms (Cialdini et al., 1990; Deutsch & Gerard, 1955), which refer to expectations of what people “should” do, independent of what behaviors that are statistically common. To ensure unfamiliar readers understand the author’s use of descriptive norms, it would help to briefly outline the contrast between descriptive and injunctive norms when explaining what the present paper aims to study.
3. Related to point 3, it would also be useful for the author to clarify in the introduction or methods whether players are explicitly aware that the other players are AI controlled. Given that injunctive norms (or normative influence) are considered by many to be driven by the expectations of other social agents (e.g. Bicchieri, 2010; Hawkins et al., 2019; Theriault et al., 2020), it seems likely that the observed effects would be stronger when participants interact with (or believe they are interacting with) real people. Does the author think this would be the case? A brief discussion of this would be useful.
4. The author writes on line 390 that “this combination [of active behavior and harmful outcome for others] seems to contribute to the unique persistence of social norms”. As written, this statement is very general and could be taken to mean that the norms we would expect to

remain most persistent are typically active and harmful. Many counterexamples can be called to mind (e.g. norms for driving on the left/right side of the road; norms for waiting one's turn to speak, etc), so if the author does believe this is true then more argumentation is necessary, as. If the author does not mean to refer to all social norms then could he clarify the intended meaning?

Minor points

5. The design of this game is unusual (and I think very interesting!) because it does not provide any monetary bonus for a high score (as far as I can tell from the manuscript). The author notes this in the discussion, and distinguishes his design from traditional economic games, but it would be helpful to point out their feature earlier in the methods or introduction as well, as most readers will assume that stars provide some material reward.

6. At line 129, the author notes that the estimated effect size is 0.5. What effect size statistic is being referred to here, and for which comparison? How was this estimated?

7. There is a typo on line 361, referring to Figure 4B, which I believe should be Figure 5B.

References

Bicchieri, C. (2010). Norms, preferences, and conditional behavior. *Politics, Philosophy & Economics*, 9(3), 297-313. <https://doi.org/10.1177/1470594X10369276>

Cialdini, R. B., Reno, R. R., & Kalgren, C. A. (1990). A focus theory of normative conduct: Recycling the concept of norms to reduce littering in public places. *Journal of Personality and Social Psychology*, 58(6), 1015-1026. <https://doi.org/10.1037/0022-3514.58.6.1015>

Deutsch, M., & Gerard, H. B. (1955). A study of normative and informational social influences upon individual judgment. *The Journal of Abnormal and Social Psychology*, 51(3), 629-636. <https://doi.org/10.1037/h0046408>

Hawkins, R. X. D., Goodman, N. D., & Goldstone, R. L. (2019). The emergence of social norms and conventions. *Trends in Cognitive Sciences*, 23(2), 158-169. <https://doi.org/10.1016/j.tics.2018.11.003>

Theriault, J. E., Young, L., & Barrett, L. F. (2020). The sense of should: A biologically-based framework for modeling social pressure. *Physics of Life Reviews*. <https://doi.org/10.1016/j.plrev.2020.01.004>

Review form: Reviewer 2

Recommendation

Major revision is needed (please make suggestions in comments)

Scientific importance: Is the manuscript an original and important contribution to its field?

Good

General interest: Is the paper of sufficient general interest?

Good

Quality of the paper: Is the overall quality of the paper suitable?

Excellent

Is the length of the paper justified?

Yes

Should the paper be seen by a specialist statistical reviewer?

Yes

Do you have any concerns about statistical analyses in this paper? If so, please specify them explicitly in your report.

Yes

It is a condition of publication that authors make their supporting data, code and materials available - either as supplementary material or hosted in an external repository. Please rate, if applicable, the supporting data on the following criteria.

Is it accessible?

Yes

Is it clear?

Yes

Is it adequate?

Yes

Do you have any ethical concerns with this paper?

No

Comments to the Author

The author investigates the cognitive mechanisms behind social norm learning. In a single study, they found that participants changed their behaviors according to various social norms of other bot-players. Behavioral analyses indicate asymmetries in learning of social norms based on experience, where learning to avoid a harmful action was unsuccessful after participants learned to engage in a harmful action. Computational models tested whether these norms were learned at the individual or group level and demonstrated that adoption of negative outcome norms was associated with group learning while positive outcome norms was associated with individual-specific learning. Taken together, the author suggests that these cognitive learning mechanisms account for adaptation to descriptive social norms.

The method that the author developed provides a rich setting to explore the development and learning of social norms. I believe the field would benefit from such dynamic and unique environments such as the Star-Harvest Game and think the work is of interest to many. I have a few suggestions for potential analyses to provide a broader understanding of the behaviors in the game as well as help resolve some outstanding issues.

Major concerns

One major concern regards an asymmetry in the social norm behavior of the bot-players and subsequent effects on the computational results. The author writes on page 8 line 163 "Benefit-action bot-players would start every turn with a probability of zapping others, even if they were closest to a star." This choice seems unusual since it makes the learning signal provided from benefit-action behavior stronger than the other norms as the opportunity cost of performing the behavior is higher (i.e., the agent not only gives up a "free" star, but they also give another player a star). This difference in signal strength makes interpreting the estimated group-level transfer effects difficult, as the benefit-action shows the least group-level transfer. It is possible that because the benefit-action bot involves an extra altruistic sacrifice that participants are more likely to engage in individual-specific learning and potentially drives the conclusion on page 17 line 404 ("Behaviours with beneficial contingency were associated with low group-level generalization, suggesting more personal and reciprocity-based learning for prosocial behaviors"). Finally, in the computational modeling results the author writes on page 12 line 268 "In all models, no learning was done if the observed player ... was the closest player to a start and moved toward it." This seems to bias learning about the benefit-action player since there

should be more opportunities to learn about them as they are the only bot to be close to a star and not move toward it. The author should present a justification for the asymmetry in norm algorithms and acknowledge this potential confound when making broad conclusions about norm learning.

Another major concern regards additional behavioral measures which could be presented. The author operationalizes the behavioral marker of adaptation to social norm as the percentage of time participants zapped when they had the opportunity to do so. When I played the game online (which I greatly appreciated thank you) I experimented and realized, at least in this demo, it was possible to zap even when the participant was not next to another bot-player in a row or column. The author should clarify zapping rules and present how often participants zapped when they did not share a column or row with another bot. Furthermore, it would be helpful both the average number of learning episodes (i.e., how often in 70 trials is there an opportunity to learn) as well as the average number of “useful” zapping opportunities from the participant (i.e., how many times could they meaningfully zap another player). Relatedly, the author should present more information about the overall zap numbers in each condition. For example, some norm dynamics which encourage more “aggressive” play might result in more opportunities to zap, which influences the learning signals and behaviors, and the raw numbers would help readers understand these dynamics.

In the author’s design there are two opportunities for learning, once in the first experimental block and once in the second. The implications about learning norms in these two circumstances are different since one represents learning in the absence of experience while the other reflects adaptation to new social norms after potentially already learning one. The behavioral results analyzed each block separately and showed no interaction between zap behavior and zap outcome in the first block, suggesting no asymmetries in learning social norms from scratch. However, the computational modeling results were fit to participants decisions across both experimental blocks which seems to ignore this ordering effect. The author should separate the computational modeling results according to order by analyzing the first and second block separately. In general, it would be helpful to clarify which results and conclusions are derived from the first versus second experimental block, since it affects generalizability of the results.

The last major concern regards how well the model accounts for the behavioral data. I appreciate the fact that the author is developing learning models in a novel environment, but having some model checks even in the Supplement would be helpful to know how best to interpret the hybrid model. For example, it would be helpful to show that the model can recover participant’s zapping behavior over time as they experience relevant learning episodes (bot-player’s zapping). Most importantly, it is critical to understand how differences in the group-level updating parameter (G_{zap} , G_{avoid}) would predict qualitatively different pattern of results seen in participant’s behavior. The author may want to consider adding learning curves demonstrating how agents who learn from the individual, group, or a hybrid of both would increase their zapping behavior over time compared to the average participant. Another way to demonstrate this behaviorally may be to show how often a participant zaps a specific bot-player (indexed by color) compared to how often they experienced that bot-player’s zap. This would assist the reader in understanding how these two different approaches to learning social norms would result in different behavior.

In a related concern, I think that the reciprocity-based model would benefit greatly from the addition of a “direct” versus “indirect” reciprocity parameter. Extensive literature has demonstrated how direct (experiencing another bot zapping the participant) and indirect (seeing the bot zap another bot) reciprocity account for different behavioral patterns (e.g., Rand & Nowak, 2013 for an overview). I believe this analysis would strengthen the results by examining whether there is a special emphasis in the learning process for observing a norm versus experiencing it.

Minor concerns:

1. Page 4, figure 1: it may be helpful to separate this figure based on the learning strategies

(individual-specific or group generalization panels) to demonstrate the authors point more clearly

2. Page 6, lines 103-105: the prediction that “behaviors with aversive outcomes may be more readily generalized to all group members than helping behaviors” is not justified based on the literature presented.
3. Page 12, line 280: The prior refers to the probability of zapping before any experience, but the model includes two experimental blocks where the norm changes. Because participants presumably learn and update their prior after the first experimental block, it may be useful to specify two priors based on the order of norm sequences the participant received as another possible control for the sequential nature of the blocks.
4. Page 14, line 315: the author may want to exclude participants who never zapped in their behavioral results as well if they are going to be removed from the modeling, because the modeling results inform the behavioral.
5. Page 14, line 319: “simple reciprocity learning rule” seems like a misnomer as the reciprocity rule is more complex than the group rule since it requires updating three separate players whereas the group rule only needs to update a single group value. At minimum the reciprocity learning rule requires more working memory and is more complex in that respect.
6. Page 17, line 390: I think the author should avoid this claim since the resistant norm being the active behavior and harmful outcome for others is likely context specific.
7. Page 18, lines 431-452: The claims about economic games only taking a “snapshot of participant’s tendencies at one point” is unfounded. Economic games are often used for both learning paradigms and dynamics of repeated play. The claims that economic games have “limited set of behaviors and norms ... focusing on ... cooperating or defecting” is unnecessary and simply untrue. There are numerous studies employing economic games to examine behaviors relating to trust, generosity, punishment, etc. The claim that this was “a social setting which participants have less experience, a video game, rather than monetary transaction tasks that are familiar” is unfounded. In general, the paragraph makes several unsubstantiated claims and framing this paragraph about the positives of the author’s paradigm may be better suited.
8. Page 19, line 458: I’m not sure what this sentence means: “a mixture of individual and group-level learning was shown to make some norms more resilient than others.” The author should clarify where this conclusion arises.

Decision letter (RSPB-2021-0293.R0)

21-Mar-2021

Dear Dr Hertz:

Your manuscript has now been peer reviewed and the reviews have been assessed by an Associate Editor. All of them, and myself, find your paradigm to be novel and interesting, with potentially very important results. However, both reviewers identify a number of issues that require further clarification, and make some recommendations for further analyses that would provide more information. I invite you to revise your manuscript to address these concerns. You will find their detailed comments appended at the end of this email.

To submit your revision please log into <http://mc.manuscriptcentral.com/prsb> and enter your Author Centre, where you will find your manuscript title listed under "Manuscripts with

Decisions." Under "Actions", click on "Create a Revision". Your manuscript number has been appended to denote a revision.

Research ethics:

Use of animals and field studies:

It is a condition of publication that you make available the data and research materials supporting the results in the article. Please see our Data Sharing Policies (<https://royalsociety.org/journals/authors/author-guidelines/#data>). Datasets should be deposited in an appropriate publicly available repository and details of the associated accession number, link or DOI to the datasets must be included in the Data Accessibility section of the article (<https://royalsociety.org/journals/ethics-policies/data-sharing-mining/>). Reference(s) to datasets should also be included in the reference list of the article with DOIs (where available).

All supplementary materials accompanying an accepted article will be treated as in their final form. They will be published alongside the paper on the journal website and posted on the online figshare repository. Files on figshare will be made available approximately one week before the

accompanying article so that the supplementary material can be attributed a unique DOI. Please try to submit all supplementary material as a single file.

Please submit a copy of your revised paper within three weeks. If we do not hear from you within this time your manuscript will be rejected. If you are unable to meet this deadline please let us know as soon as possible, as we may be able to grant a short extension.

Best wishes,
Dr Sarah Brosnan
Editor, Proceedings B
mailto:proceedingsb@royalsociety.org

Associate Editor
Board Member: 1
Comments to Author:

Two reviewers have provided thoughtful feedback on this article. Both praised the novelty of the research paradigm and the questions posed. However, both reviewers also provide suggestions as to how the reporting could be clarified and strengthened, including additional analysis of the data set to offer a more nuanced understanding of the players' responses and a more thorough establishment of the models. These edits, in addition to some refinement of the framing and language throughout, will help to strengthen and deepen the insights offered from this study and findings.

Reviewer(s)' Comments to Author:

Referee: 1

Comments to the Author(s)

This paper presents data collected in an online game alongside a computational model to demonstrate that influence from group-level behavior affects the behavior of participants above and beyond their interactions with other individual agents (e.g. in reciprocal exchanges of rewards/punishments). Players controlled and agent and navigated in a 2D space, collecting star rewards and delivering "zaps" that had either positive or negative effects on AI controlled agents. I really enjoyed this paper and believe the models and the method can contribute to our understanding of norms. I believe it would be a strong publication in Proceedings of the Royal Academy B, as long as a few points can be clarified and/or elaborated on.

1. Most significantly, it would help to have a stronger test of agent-based influence included in the author's computational model. His model of reciprocity-based learning appears to model the influence of AI agent's "zapping" on the participant's "zapping" behavior by observing the AI's use of "zaps", but the distinction between second-party and third-party reward/punishment is not discussed. Would it be possible to separately model the influence of "zaps" that players personally receive from other agents (i.e. second-party reciprocity), while separately modeling the influence of "zaps" that players observe between the bots (i.e. third-party reciprocity)? It seems likely that players are more likely to reciprocate "zaps" to individual bots when they were directly affected by them, and modeling this make improve the fit of the reciprocity-based model.

2. The author describes his study as examining descriptive norms, which I agree is correct. However, descriptive norms are usually described in contrast to injunctive norms (Cialdini et al., 1990; Deutsch & Gerard, 1955), which refer to expectations of what people “should” do, independent of what behaviors that are statistically common. To ensure unfamiliar readers understand the author’s use of descriptive norms, it would help to briefly outline the contrast between descriptive and injunctive norms when explaining what the present paper aims to study.
3. Related to point 3, it would also be useful for the author to clarify in the introduction or methods whether players are explicitly aware that the other players are AI controlled. Given that injunctive norms (or normative influence) are considered by many to be driven by the expectations of other social agents (e.g. Bicchieri, 2010; Hawkins et al., 2019; Theriault et al., 2020), it seems likely that the observed effects would be stronger when participants interact with (or believe they are interacting with) real people. Does the author think this would be the case? A brief discussion of this would be useful.
4. The author writes on line 390 that “this combination [of active behavior and harmful outcome for others] seems to contribute to the unique persistence of social norms”. As written, this statement is very general and could be taken to mean that the norms we would expect to remain most persistent are typically active and harmful. Many counterexamples can be called to mind (e.g. norms for driving on the left/right side of the road; norms for waiting one’s turn to speak, etc), so if the author does believe this is true then more argumentation is necessary, as. If the author does not mean to refer to all social norms then could he clarify the intended meaning?

Minor points

5. The design of this game is unusual (and I think very interesting!) because it does not provide any monetary bonus for a high score (as far as I can tell from the manuscript). The author notes this in the discussion, and distinguishes his design from traditional economic games, but it would be helpful to point out their feature earlier in the methods or introduction as well, as most readers will assume that stars provide some material reward.
6. At line 129, the author notes that the estimated effect size is 0.5. What effect size statistic is being referred to here, and for which comparison? How was this estimated?
7. There is a typo on line 361, referring to Figure 4B, which I believe should be Figure 5B.

References

- Bicchieri, C. (2010). Norms, preferences, and conditional behavior. *Politics, Philosophy & Economics*, 9(3), 297–313. <https://doi.org/10.1177/1470594X10369276>
- Cialdini, R. B., Reno, R. R., & Kalgren, C. A. (1990). A focus theory of normative conduct: Recycling the concept of norms to reduce littering in public places. *Journal of Personality and Social Psychology*, 58(6), 1015–1026. <https://doi.org/10.1037/0022-3514.58.6.1015>
- Deutsch, M., & Gerard, H. B. (1955). A study of normative and informational social influences upon individual judgment. *The Journal of Abnormal and Social Psychology*, 51(3), 629–636. <https://doi.org/10.1037/h0046408>
- Hawkins, R. X. D., Goodman, N. D., & Goldstone, R. L. (2019). The emergence of social norms and conventions. *Trends in Cognitive Sciences*, 23(2), 158–169. <https://doi.org/10.1016/j.tics.2018.11.003>
- Theriault, J. E., Young, L., & Barrett, L. F. (2020). The sense of should: A biologically-based framework for modeling social pressure. *Physics of Life Reviews*. <https://doi.org/10.1016/j.plrev.2020.01.004>

Referee: 2

Comments to the Author(s)

The author investigates the cognitive mechanisms behind social norm learning. In a single study, they found that participants changed their behaviors according to various social norms of other

bot-players. Behavioral analyses indicate asymmetries in learning of social norms based on experience, where learning to avoid a harmful action was unsuccessful after participants learned to engage in a harmful action. Computational models tested whether these norms were learned at the individual or group level and demonstrated that adoption of negative outcome norms was associated with group learning while positive outcome norms was associated with individual-specific learning. Taken together, the author suggests that these cognitive learning mechanisms account for adaptation to descriptive social norms.

The method that the author developed provides a rich setting to explore the development and learning of social norms. I believe the field would benefit from such dynamic and unique environments such as the Star-Harvest Game and think the work is of interest to many. I have a few suggestions for potential analyses to provide a broader understanding of the behaviors in the game as well as help resolve some outstanding issues.

Major concerns

One major concern regards an asymmetry in the social norm behavior of the bot-players and subsequent effects on the computational results. The author writes on page 8 line 163 "Benefit-action bot-players would start every turn with a probability of zapping others, even if they were closest to a star." This choice seems unusual since it makes the learning signal provided from benefit-action behavior stronger than the other norms as the opportunity cost of performing the behavior is higher (i.e., the agent not only gives up a "free" star, but they also give another player a star). This difference in signal strength makes interpreting the estimated group-level transfer effects difficult, as the benefit-action shows the least group-level transfer. It is possible that because the benefit-action bot involves an extra altruistic sacrifice that participants are more likely to engage in individual-specific learning and potentially drives the conclusion on page 17 line 404 ("Behaviours with beneficial contingency were associated with low group-level generalization, suggesting more personal and reciprocity-based learning for prosocial behaviors"). Finally, in the computational modeling results the author writes on page 12 line 268 "In all models, no learning was done if the observed player ... was the closest player to a start and moved toward it." This seems to bias learning about the benefit-action player since there should be more opportunities to learn about them as they are the only bot to be close to a star and not move toward it. The author should present a justification for the asymmetry in norm algorithms and acknowledge this potential confound when making broad conclusions about norm learning.

Another major concern regards additional behavioral measures which could be presented. The author operationalizes the behavioral marker of adaptation to social norm as the percentage of time participants zapped when they had the opportunity to do so. When I played the game online (which I greatly appreciated thank you) I experimented and realized, at least in this demo, it was possible to zap even when the participant was not next to another bot-player in a row or column. The author should clarify zapping rules and present how often participants zapped when they did not share a column or row with another bot. Furthermore, it would be helpful both the average number of learning episodes (i.e., how often in 70 trials is there an opportunity to learn) as well as the average number of "useful" zapping opportunities from the participant (i.e., how many times could they meaningfully zap another player). Relatedly, the author should present more information about the overall zap numbers in each condition. For example, some norm dynamics which encourage more "aggressive" play might result in more opportunities to zap, which influences the learning signals and behaviors, and the raw numbers would help readers understand these dynamics.

In the author's design there are two opportunities for learning, once in the first experimental block and once in the second. The implications about learning norms in these two circumstances are different since one represents learning in the absence of experience while the other reflects adaptation to new social norms after potentially already learning one. The behavioral results analyzed each block separately and showed no interaction between zap behavior and zap outcome in the first block, suggesting no asymmetries in learning social norms from scratch.

However, the computational modeling results were fit to participants decisions across both experimental blocks which seems to ignore this ordering effect. The author should separate the computational modeling results according to order by analyzing the first and second block separately. In general, it would be helpful to clarify which results and conclusions are derived from the first versus second experimental block, since it affects generalizability of the results.

The last major concern regards how well the model accounts for the behavioral data. I appreciate the fact that the author is developing learning models in a novel environment, but having some model checks even in the Supplement would be helpful to know how best to interpret the hybrid model. For example, it would be helpful to show that the model can recover participant's zapping behavior over time as they experience relevant learning episodes (bot-player's zapping). Most importantly, it is critical to understand how differences in the group-level updating parameter (G_{zap} , G_{avoid}) would predict qualitatively different pattern of results seen in participant's behavior. The author may want to consider adding learning curves demonstrating how agents who learn from the individual, group, or a hybrid of both would increase their zapping behavior over time compared to the average participant. Another way to demonstrate this behaviorally may be to show how often a participant zaps a specific bot-player (indexed by color) compared to how often they experienced that bot-player's zap. This would assist the reader in understanding how these two different approaches to learning social norms would result in different behavior.

In a related concern, I think that the reciprocity-based model would benefit greatly from the addition of a "direct" versus "indirect" reciprocity parameter. Extensive literature has demonstrated how direct (experiencing another bot zapping the participant) and indirect (seeing the bot zap another bot) reciprocity account for different behavioral patterns (e.g., Rand & Nowak, 2013 for an overview). I believe this analysis would strengthen the results by examining whether there is a special emphasis in the learning process for observing a norm versus experiencing it.

Minor concerns:

1. Page 4, figure 1: it may be helpful to separate this figure based on the learning strategies (individual-specific or group generalization panels) to demonstrate the authors point more clearly
2. Page 6, lines 103-105: the prediction that "behaviors with aversive outcomes may be more readily generalized to all group members than helping behaviors" is not justified based on the literature presented.
3. Page 12, line 280: The prior refers to the probability of zapping before any experience, but the model includes two experimental blocks where the norm changes. Because participants presumably learn and update their prior after the first experimental block, it may be useful to specify two priors based on the order of norm sequences the participant received as another possible control for the sequential nature of the blocks.
4. Page 14, line 315: the author may want to exclude participants who never zapped in their behavioral results as well if they are going to be removed from the modeling, because the modeling results inform the behavioral.
5. Page 14, line 319: "simple reciprocity learning rule" seems like a misnomer as the reciprocity rule is more complex than the group rule since it requires updating three separate players whereas the group rule only needs to update a single group value. At minimum the reciprocity learning rule requires more working memory and is more complex is that respect.
6. Page 17, line 390: I think the author should avoid this claim since the resistant norm being the active behavior and harmful outcome for others is likely context specific.
7. Page 18, lines 431-452: The claims about economic games only taking a "snapshot of participant's tendencies at one point" is unfounded. Economic games are often used for both learning paradigms and dynamics of repeated play. The claims that economic games have "limited set of behaviors and norms ... focusing on ... cooperating or defecting" is unnecessary and simply untrue. There are numerous studies employing economic games to examine behaviors relating to trust, generosity, punishment, etc. The claim that this was "a social setting which participants have less experience, a video game, rather than monetary transaction tasks

that are familiar” is unfounded. In general, the paragraph makes several unsubstantiated claims and framing this paragraph about the positives of the author’s paradigm may be better suited.

8. Page 19, line 458: I’m not sure what this sentence means: “a mixture of individual and group-level learning was shown to make some norms more resilient than others.” The author should clarify where this conclusion arises.

Author's Response to Decision Letter for (RSPB-2021-0293.R0)

See Appendix A.

RSPB-2021-0293.R1 (Revision)

Review form: Reviewer 1

Recommendation

Accept as is

Scientific importance: Is the manuscript an original and important contribution to its field?

Excellent

General interest: Is the paper of sufficient general interest?

Excellent

Quality of the paper: Is the overall quality of the paper suitable?

Excellent

Is the length of the paper justified?

Yes

Should the paper be seen by a specialist statistical reviewer?

No

Do you have any concerns about statistical analyses in this paper? If so, please specify them explicitly in your report.

No

It is a condition of publication that authors make their supporting data, code and materials available - either as supplementary material or hosted in an external repository. Please rate, if applicable, the supporting data on the following criteria.

Is it accessible?

Yes

Is it clear?

Yes

Is it adequate?

Yes

Do you have any ethical concerns with this paper?

No

Comments to the Author

The author ought to be praised for the careful attention to detail in their response to reviews. I felt that all my original concerns were appropriately addressed and appreciate the thoughtfulness of the author.

After re-reading the manuscript, I have no outstanding concerns.

Review form: Reviewer 2

Recommendation

Accept as is

Scientific importance: Is the manuscript an original and important contribution to its field?

Excellent

General interest: Is the paper of sufficient general interest?

Good

Quality of the paper: Is the overall quality of the paper suitable?

Excellent

Is the length of the paper justified?

Yes

Should the paper be seen by a specialist statistical reviewer?

No

Do you have any concerns about statistical analyses in this paper? If so, please specify them explicitly in your report.

No

It is a condition of publication that authors make their supporting data, code and materials available - either as supplementary material or hosted in an external repository. Please rate, if applicable, the supporting data on the following criteria.

Is it accessible?

Yes

Is it clear?

Yes

Is it adequate?

Yes

Do you have any ethical concerns with this paper?

No

Comments to the Author

Thank you for your detailed response. All of my concerns have been addressed.

Decision letter (RSPB-2021-0293.R1)

10-May-2021

Dear Dr Hertz

I am pleased to inform you that your manuscript entitled "Learning how to behave: Cognitive learning processes account for asymmetries in adaptation to social norms" has been accepted for publication in Proceedings B. I also wish to add that your responses to the reviewers were particularly thorough and detailed, which we all really appreciated. I look forward to your next work with this system.

Data Accessibility section

Open Access

You are invited to opt for Open Access, making your freely available to all as soon as it is ready for publication under a CCBY licence. Our article processing charge for Open Access is £1700. Corresponding authors from member institutions (<http://royalsocietypublishing.org/site/librarians/allmembers.xhtml>) receive a 25% discount to these charges. For more information please visit <http://royalsocietypublishing.org/open-access>.

Your article has been estimated as being 9 pages long. Our Production Office will be able to confirm the exact length at proof stage.

Paper charges

Sincerely,

Dr Sarah Brosnan

Associate Editor:

Board Member: 1

Comments to Author:

Thank you for offering such detailed and thoughtful responses to the reviewers' feedback. Both reviewers agree that you have thoroughly addressed their concerns and responded to their comments with care.

Appendix A

Dear Editors and reviewers,

Thank you for the opportunity to revise my paper, and for the useful comments and the appreciation of this project. As the reviewers suggested, additional analyses and descriptions of participants' behaviour are included in the revised version, as well as more detailed descriptions of the models. In addition, I revised some parts of the theoretical framework and discussed the limitations of this work more thoroughly.

I also discussed in the response letter the effect of direct and indirect reciprocity in this task. I think that this is a very interesting question, and one that I am currently working on in follow up projects. While the notion of direct and indirect reciprocity is relevant to the problem of learning social norms, I think it goes beyond the scope of the current work, which was not optimized to study this problem, and that this subject is better off addressed in a work dedicated to it. I discuss these limitations in detail and give two examples of follow-up studies that I am currently carrying, but I did not include most of these discussions and new data in the revised manuscript.

Below are point-by-point responses to the reviewers' comments. I hope that you will find my responses satisfying.

Best wishes,

Uri Hertz

Associate Editor

Board Member: 1

Comments to Author:

Two reviewers have provided thoughtful feedback on this article. Both praised the novelty of the research paradigm and the questions posed. However, both reviewers also provide suggestions as to how the reporting could be clarified and strengthened, including additional analysis of the data set to offer a more nuanced understanding of the players' responses and a more thorough establishment of the models. These edits, in addition to some refinement of the framing and language throughout, will help to strengthen and deepen the insights offered from this study and findings.

Reviewer(s)' Comments to Author:

Referee: 1

Comments to the Author(s)

This paper presents data collected in an online game alongside a computational model to demonstrate that influence from group-level behavior affects the behavior of participants above and beyond their interactions with other individual agents (e.g. in

reciprocal exchanges of rewards/punishments). Players controlled and agent and navigated in a 2D space, collecting star rewards and delivering “zaps” that had either positive or negative effects on AI controlled agents. I really enjoyed this paper and believe the models and the method can contribute to our understanding of norms. I believe it would be a strong publication in Proceedings of the Royal Academy B, as long as a few points can be clarified and/or elaborated on.

1. Most significantly, it would help to have a stronger test of agent-based influence included in the author’s computational model. His model of reciprocity-based learning appears to model the influence of AI agent’s “zapping” on the participant’s “zapping” behavior by observing the AI’s use of “zaps”, but the distinction between second-party and third-party reward/punishment is not discussed. Would it be possible to separately model the influence of “zaps” that players personally receive from other agents (i.e. second-party reciprocity), while separately modeling the influence of “zaps” that players observe between the bots (i.e. third-party reciprocity)? It seems likely that players are more likely to reciprocate “zaps” to individual bots when they were directly affected by them, and modeling this make improve the fit of the reciprocity-based model.

A1:

Thank you for this comment, which was also raised by reviewer 2. The answers to both comments is therefore essentially the same.

Examining the different influences of first-hand experience and second-hand observation on learning is a very good suggestion. However, as I will show below, it may be a bit more complicated than simply expanding the computational models with the existing experimental design. The current experimental design, and indeed the main focus of the paper, have to do with the effect of behavioural prescription and experience on learning of social norms, while the specific group structure (who did what to whom) was not manipulated. This means that all bot-players had similar probabilities to zap others and zap the player. While it is possible to examine how the participants’ zapping behaviour was related to first and second-hand experiences, it may not be very informative in the current design. I am currently running two follow-up experiments looking more closely at reciprocity and the role of first and second-hand information, which are based on the current experimental design and findings (see below).

To address the comment raised by the reviewer, I first expanded the hybrid learning model used in the manuscript to include different learning rates for experienced and observed zaps and avoidances:

First-hand update rule:

$$ZapProb_{t+1}^p = ZapProb_t^p + \begin{cases} LR_{Zap}^{Experience} \cdot (1 - ZapProb_t^p) & Z_t = 1 \\ LR_{Avoid}^{Experience} \cdot (0 - ZapProb_t^p) & Z_t = 0 \end{cases}$$

Second-hand update rule:

$$ZapProb_{t+1}^p = ZapProb_t^p + \begin{cases} LR_{Zap}^{Observe} \cdot (1 - ZapProb_t^p) & Z_t = 1 \\ LR_{Avoid}^{Observe} \cdot (0 - ZapProb_t^p) & Z_t = 0 \end{cases}$$

With the same learning rules used to generalize to other players, using the generalization parameters as in the hybrid model. I used a similar fitting procedure as the one used for the models in the main text. This model had a higher average DIC score in the negative zap conditions ($DIC_{New} - DIC_{Hybrid} = 6.6$, $t(129) = 14.28$, $p < 0.0001$), and was comparable with the hybrid model in the positive zap conditions ($DIC_{New} - DIC_{Hybrid} = -0.09$, $t(112) = -0.91$, $p = 0.36$). These results indicate that the addition of parameters aimed to account for first and second-hand learning did not increase the model fitting to the data sufficiently.

A description of the additional computational model and results is now included in the supplementary materials.

One possible cause for the reduced performance of this model may have to do with the current experimental design. Currently, participants experienced two sets of bot-players, one which did not zap at all and another that zapped from time to time. The models have to account for both sets of participants, and to the adaptation in behaviour when moving from one environment to another. If people behave differently within one environment than between environments, this model will be likely to underperform.

For example, in the current experiment participants were found to overall zap all bot-players with similar frequency, regardless of how much these players zapped overall in the harm-active conditions (green bars in figure Rev1). When moving to the new environment, where bot-players did not zap, zapping behaviour could not be associated with individual bot-player zaps, as there were no such zaps. This difference between conditions is accounted for by the models in this paper and is indeed the focus of this project.

However, when breaking the bot-players' zaps to first and second-hand zaps (yellow and blue bars respectively in the figure Rev1), a more complicated picture emerges. It seems that participants were more likely to zap players that zapped them the least, and most likely to zap players that zapped others the most. First and second-hand learning therefore affected behaviour differently in our task. Such within block differences are hard to account for in the current design.

Figure Rev1: Participants' zapping behaviour towards specific bot-player, according to their overall number of zaps (Green), number of zaps of the participant (Yellow), and number of zaps of other bot-players (Blue).

To demonstrate that unpacking this pattern may be beyond the scope of the current project, I added results from two follow-up experiments.

In figure Rev2, I present results from a follow-up experiment in which participants were playing with three bot-players displaying different zapping behaviours. Two bot-players were zappers, following the harm-action norm, and one was non-zapper, following a harm-avoidance pattern. It is clear from this figure that the fact that the harm-avoider did not zap even once was registered by the participants, and they tended to zap this player less than others. This can be seen in the first-hand zapping behaviour, showing the opposite pattern of what was observed when the player that zapped the participant the least (but at least once, in the current study) was zapped the most (in figure Rev1).

Figure Rev2: Results from follow-up experiment 1, where one bot-player(Min) follows a harm-avoidance pattern, while the two other players display a harm-action pattern.

A second follow-up study was designed to examine whether first-hand experiences are crucial for behavioural adaptation. In this study participants played with bot-players that displayed different behaviours toward each other, and toward the participant. They either zapped each other and avoided zapping the participant, or zapped the participant and avoided zapping each other. The results indicate that participants' behaviour was dependent on their first-hand experience – they zapped more when they were being zapped and avoided zapping when the players avoided zapping them (Figure Rev3, pink and purple bars).

Figure Rev3: Results from follow-up experiment 2, where the bot-players displayed different behaviour amongst themselves than toward the participant.

Taken together, these follow-up studies indicate that homogeneity of the norm behaviour, i.e. being displayed by most group members most of the time (Ullmann-Margalit, 2015), is important in the learning and adaptation of social norms.

The current work examined how behavioural features of social norms affect learning and adaptation, and therefore behaviours were displayed uniformly by the players. A more refined examination of the dependencies of social learning on the specific pattern of displayed behaviour (who do what to whom) could not be carried directly in the current settings. Using the same experimental framework, it is possible to examine more intricate social structure and dynamics, as was demonstrated in the follow-up studies.

A discussion on the limitation of current design in the study of more refined social learning strategies, and outline of future direction, was added to the discussion. The detailed description of the follow-up studies and the extra figures are not included in the revised manuscript or the supplementary materials.

Finally, I changed the name of the *Reciprocity* model to *Individual* model. This was done to highlight the main feature of the model, which is learning on individual level with no generalization to other players, but to avoid confusion with reciprocity as a first-hand experience.

2. The author describes his study as examining descriptive norms, which I agree is correct. However, descriptive norms are usually described in contrast to injunctive norms (Cialdini et al., 1990; Deutsch & Gerard, 1955), which refer to expectations of what people “should” do, independent of what behaviors that are statistically common. To ensure unfamiliar readers understand the author’s use of descriptive norms, it would help to briefly outline the contrast between descriptive and injunctive norms when explaining what the present paper aims to study.

A2:

Thank you for this comment, a short description of injunctive norms was added to the introduction.

3. Related to point 3, it would also be useful for the author to clarify in the introduction or methods whether players are explicitly aware that the other players are AI controlled. Given that injunctive norms (or normative influence) are considered by many to be driven by the expectations of other social agents (e.g. Bicchieri, 2010; Hawkins et al., 2019; Theriault et al., 2020), it seems likely that the observed effects would be stronger when participants interact with (or believe they are interacting with) real people. Does the author think this would be the case? A brief discussion of this would be useful.

A3:

Participants were not given explicit information regarding the identity of the other players, either if these were humans or bots, and participants were simply told that they will play the game with other players, indicated by different colours in the star-harvest task. As the experiments were taking place online, where it is both possible to interact anonymously with other humans and where some interactions are with algorithmic bots, supported this ambiguity.

It is possible that during interactions with humans the patterns observed here will be amplified or different. However, participants’ behaviour in the positive-zaps conditions, where zaps were mainly to benefit others and had no clear benefit for the participant, suggests that participants did treat the other players as if they were fellow participants, to some extent. Participants tended to zap the other players more in the benefit-action norm than in the benefit-omission norm, suggesting that, to some extent, people played as if the other players were humans and they had some reputational or pro-social incentives.

A short section describing these concerns is now added to the discussion.

4. The author writes on line 390 that “this combination [of active behavior and harmful outcome for others] seems to contribute to the unique persistence of social norms”. As written, this statement is very general and could be taken to mean that the norms we would expect to remain most persistent are typically active and harmful. Many counterexamples can be called to mind (e.g. norms for driving on the left/right side of the road; norms for waiting one’s turn to speak, etc), so if the author does believe this is true then more argumentation is necessary, as. If the author does not mean to refer to all social norms then could he clarify the intended meaning?

A4:

This statement is now refined to reflect that this is the combination of active behaviour and harmful outcome was found to contribute uniquely in the social norms examined in this study. However, I further suggest that this combination can be a factor in making real-life social norms persistent, and should be considered among other mechanisms, for example the way such norms are imposed and maintained by social institutions (formal and informal), social signaling role of following norms and habits.

Minor points

5. The design of this game is unusual (and I think very interesting!) because it does not provide any monetary bonus for a high score (as far as I can tell from the manuscript). The author notes this in the discussion, and distinguishes his design from traditional economic games, but it would be helpful to point out their feature earlier in the methods or introduction as well, as most readers will assume that stars provide some material reward.

A5:

The dissociation of monetary reward from performance in the task is now explicitly mentioned in the methods and descriptions of the task and in the discussion.

6. At line 129, the author notes that the estimated effect size is 0.5. What effect size statistic is being referred to here, and for which comparison? How was this estimated?

A6:

The effect is referring to the expected within-subject difference in average zapping between blocks, based on a pilot with transition from harm-omission to harm-action norm. This clarification was added to the text.

7. There is a typo on line 361, referring to Figure 4B, which I believe should be Figure 5B.

A7: Thank you for noticing, I corrected the figure reference.

References

Bicchieri, C. (2010). Norms, preferences, and conditional behavior. *Politics, Philosophy & Economics*, 9(3), 297–313. <https://doi.org/10.1177/1470594X10369276>

Cialdini, R. B., Reno, R. R., & Kalgren, C. A. (1990). A focus theory of normative conduct: Recycling the concept of norms to reduce littering in public places. *Journal of Personality and Social Psychology*, 58(6), 1015–1026. <https://doi.org/10.1037/0022-3514.58.6.1015>

Deutsch, M., & Gerard, H. B. (1955). A study of normative and informational social influences upon individual judgment. *The Journal of Abnormal and Social Psychology*, 51(3), 629–636. <https://doi.org/10.1037/h0046408>

Hawkins, R. X. D., Goodman, N. D., & Goldstone, R. L. (2019). The emergence of social norms and conventions. *Trends in Cognitive Sciences*, 23(2), 158–169. <https://doi.org/10.1016/j.tics.2018.11.003>

Theriault, J. E., Young, L., & Barrett, L. F. (2020). The sense of should: A biologically-based framework for modeling social pressure. *Physics of Life Reviews*. <https://doi.org/10.1016/j.plrev.2020.01.004>

Referee: 2

Comments to the Author(s)

The author investigates the cognitive mechanisms behind social norm learning. In a single study, they found that participants changed their behaviors according to various social norms of other bot-players. Behavioral analyses indicate asymmetries in learning of social norms based on experience, where learning to avoid a harmful action was unsuccessful after participants learned to engage in a harmful action. Computational models tested whether these norms were learned at the individual or group level and demonstrated that adoption of negative outcome norms was associated with group learning while positive outcome norms was associated with individual-specific learning. Taken together, the author suggests that these cognitive learning mechanisms account for adaptation to descriptive social norms.

The method that the author developed provides a rich setting to explore the development and learning of social norms. I believe the field would benefit from such dynamic and unique environments such as the Star-Harvest Game and think the work is of interest to many. I have a few suggestions for potential analyses to provide a broader understanding of the behaviors in the game as well as help resolve some outstanding issues.

Major concerns

1. One major concern regards an asymmetry in the social norm behavior of the bot-players and subsequent effects on the computational results. The author writes on page 8 line 163 “Benefit-action bot-players would start every turn with a probability of zapping others, even if they were closest to a star.” This choice seems unusual since it makes the learning signal provided from benefit-action behavior stronger than the other norms as the opportunity cost of performing the behavior is higher (i.e., the agent not only gives up a “free” star, but they also give another player a star). This difference in signal strength makes interpreting the estimated group-level transfer effects difficult, as the benefit-action shows the least group-level transfer. It is possible that because the benefit-action bot involves an extra altruistic sacrifice that participants are more likely to engage in individual-specific learning and potentially drives the conclusion on page 17 line 404 (“Behaviours with beneficial contingency were associated with low group-level generalization, suggesting more personal and reciprocity-based learning for prosocial behaviors”). Finally, in the computational modeling results the author writes on page 12 line 268 “In all models, no learning was done if the observed player ... was the closest player to a star and moved toward it.” This seems to bias learning about the benefit-action player since there should be more opportunities to learn about them as they are the only bot to be close to a star and not move toward it. The author should present a justification for the asymmetry in norm algorithms and acknowledge this potential confound when making broad conclusions about norm learning.

A1:

Thank you for raising this comment. You are correct in pointing out that the Benefit-Action norm is unique in that it allows bot-players to zap even when they are closest to a star. However, the likelihood of zapping in the Benefit-Action norm was dependent on the bot-player’s distance from a star – the likelihood to zap was the minimum between 0.5 and $(\text{distance from a star})^{1.5}/50$, which meant that probability of zapping was very low if the player was very close to a star (if he is right next to a star this probability was 0.02, when he was 3 moves away from a star the probability grow to 0.1, and it reaches 0.5 when the closest star is 9 steps away). This was mentioned in the supplementary materials, but not explicitly in the main text. I now added this information to the main text.

The reason for this distance based rule was to increase the rates of zapping in the positive condition. When developing the game, I used a deterministic rule where players first moved to the closest star, and if they were not closest to a star and could zap other players they would zap. This resulted in few zaps when stars were around, and a constant loop of zaps between pairs of players when they were not closest to a star.

Introducing a probability of zapping allowed players to zap even when there were stars around, and avoid zapping even when there no stars around. Linking this probability to the player's distance from a star made players more likely to move towards stars when they are close to them, and more likely to zap others when they are far from every star. This also resulted in very few zaps that occurred when the bot-player was closest to a star, and usually when their closest star was a couple of moves away. Importantly, as stars disappeared after a number of trials, and did not remain in place until picked, not moving towards a far star was not such a huge sacrifice, as the stars may disappear until the player reach it. Explanation of this rationale is now added to the supplementary materials.

Finally, the model fitting procedure included all zaps, and the removal of trials in which the player was closest to a star was only for avoidances. I rephrased this description in the text.

2. Another major concern regards additional behavioral measures which could be presented. The author operationalizes the behavioral marker of adaptation to social norm as the percentage of time participants zapped when they had the opportunity to do so. When I played the game online (which I greatly appreciated thank you) I experimented and realized, at least in this demo, it was possible to zap even when the participant was not next to another bot-player in a row or column. The author should clarify zapping rules and present how often participants zapped when they did not share a column or row with another bot. Furthermore, it would be helpful both the average number of learning episodes (i.e., how often in 70 trials is there an opportunity to learn) as well as the average number of "useful" zapping opportunities from the participant (i.e., how many times could they meaningfully zap another player). Relatedly, the author should present more information about the overall zap numbers in each condition. For example, some norm dynamics which encourage more "aggressive" play might result in more opportunities to zap, which influences the learning signals and behaviors, and the raw numbers would help readers understand these dynamics.

A2:

The additional behavioural measures were added to the supplementary materials, and displayed here as well.

Participants could indeed zap even when they did not share a row or a column with another player, i.e. zap at no one. The instructions did not specify when participants should zap others, but only what will happen to players affected by zaps. The summary of zap behaviour in the different experimental conditions – the overall number of zaps, number of targeted zaps and number of free-zaps – is presented in Figure Rev4 and the table below. While the main analysis in the manuscript was carried on normalized zaps, i.e. percent of targeted zaps out of all zap opportunities, the pattern of the overall number of zaps and targeted zaps is relatively similar to the normalized data. The main difference is that participants in the positive zap condition carried more free zaps than those in the negative zap conditions.

Figure Rev4: Zaps in different experimental conditions. The total number of zaps made by the participants (A), included zaps targeted at other players (B) and zaps that did not affect any other player (free-zaps, C). The number of free zaps was relatively low in the negative zaps conditions (yellow and orange) but was higher in the positive zaps conditions (green and blue).

	Harm-Avoid (1)	Harm-Action (2)	Harm-Action (1)	Harm-Avoid (2)	Benefit-Avoid (1)	Benefit-Action (2)	Benefit-Action (1)	Benefit-Avoid (2)
Overall Zaps	2.95 ± 0.14	4.43 ± 0.18	5.22 ± 0.16	4.44 ± 0.19	2.11 ± 0.11	4.68 ± 0.27	4.94 ± 0.32	2.53 ± 0.3
Targeted Zaps	2.22 ± 0.12	4 ± 0.18	4.11 ± 0.14	3.93 ± 0.17	0.69 ± 0.05	2.79 ± 0.20	2.37 ± 0.12	1.22 ± 0.08
Free Zaps	0.73 ± 0.05	0.43 ± 0.02	1.10 ± 0.08	0.51 ± 0.05	1.41 ± 0.08	1.88 ± 0.12	2.56 ± 0.29	1.3 ± 0.24

Table Rev1 – mean and SEM of zaps

The number of zap opportunities is summarized and presented in Figure Rev5A, and Table Rev2. The number of trials in which participants could carry a targeted zap, i.e. shared a row or a column with another player, was relatively similar across conditions. Participants had more zap opportunities in the benefit-action conditions, as in these conditions participants that were zapped in the previous trial stayed in the same place (unlike in the harm-action condition) and could therefore zap the player back.

Learning opportunities were trials where the bot-players either zapped someone or avoided zapping. Zap avoidance trials were trials in which a player could zap someone, was not closest to a star and did not zap. These Learning opportunities were used in the learning models to update the likelihood of players to zap. Note that overall there were 210 moves made by the three bot-players in each block. The summary of learning opportunities is presented in Figure Rev5B and Table Rev2. Participants (and models) had more learning opportunities in harm-action and benefit-action conditions, which included both zaps and avoidances. In addition, learning opportunities were symmetric, i.e. did not change according to blocks' order.

Figure Rev5: Zap opportunities and learning opportunities. (A) The number of trials out of 70 where participants had the opportunity to zap a player, i.e. they shared a column or a row. Benefit-action conditions had a higher number of zap opportunities compared to other conditions, as after being zapped participants had the opportunity to zap back, which they did not have in the zap-action condition. (B) Learning opportunities were times where the participants either experienced or observed a zap, or experienced or observed an avoidance behaviour. Avoidances were trials in which a player was not closest to a star, had an opportunity to zap another player, and did not zap. Conditions including zaps had more learning opportunities than those that did not include zaps.

	Harm-Avoid (1)	Harm-Action (2)	Harm-Action (1)	Harm-Avoid (2)	Benefit-Avoid (1)	Benefit-Action (2)	Benefit-Action (1)	Benefit-Avoid (2)
Zap Opportunities	24.7 ± 0.28	23.94 ± 0.3	23.96 ± 0.25	22.5 ± 0.28	24.09 ± 0.25	26.95 ± 0.35	27.19 ± 0.30	24.75 ± 0.25
Learning Opportunities	45.92 ± 0.44	60.56 ± 0.44	58.56 ± 0.40	47.09 ± 0.43	47.06 ± 0.36	56.95 ± 0.42	56.89 ± 0.47	48.28 ± 0.38

Table Rev2 – mean and SEM of opportunities

3. In the author's design there are two opportunities for learning, once in the first experimental block and once in the second. The implications about learning norms in these two circumstances are different since one represents learning in the absence of experience while the other reflects adaptation to new social norms after potentially already learning one. The behavioral results analyzed each block separately and showed no interaction between zap behavior and zap outcome in the first block, suggesting no asymmetries in learning social norms from scratch. However, the computational modeling results were fit to participants decisions across both experimental blocks which seems to ignore this ordering effect. The author should separate the computational modeling results according to order by analyzing the first and second block separately. In general, it would be helpful to clarify which results and

conclusions are derived from the first versus second experimental block, since it affects generalizability of the results.

A3:

The reviewer is correct in the description of the task, and the analysis of the first block. However, another behavioural analysis examined adaptation between the two blocks – i.e. did participants zap more when moving from the first block to the second. This analysis was carried on the difference in zap percentages between the action and omission blocks for each participant, and its dependent variables were blocks order (action->omission or omission->action), and zap outcome (benefit/harm). The results of this analysis revealed the asymmetry in adaptation, where participants moving from harm-action block to harm-omission block showed reduced adaptation. This was demonstrated in the differences in bars in Figure 3. This analysis, therefore, included both blocks and was sensitive to blocks order. The computational models were aimed at examining learning mechanisms underlying the observed asymmetry in adaptation, and therefore included data from both blocks, and were sensitive to order as well.

I now highlight these features in the results section.

4. The last major concern regards how well the model accounts for the behavioral data. I appreciate the fact that the author is developing learning models in a novel environment, but having some model checks even in the Supplement would be helpful to know how best to interpret the hybrid model. For example, it would be helpful to show that the model can recover participant's zapping behavior over time as they experience relevant learning episodes (bot-player's zapping). Most importantly, it is critical to understand how differences in the group-level updating parameter (G_{zap} , G_{avoid}) would predict qualitatively different pattern of results seen in participant's behavior. The author may want to consider adding learning curves demonstrating how agents who learn from the individual, group, or a hybrid of both would increase their zapping behavior over time compared to the average participant. Another way to demonstrate this behaviorally may be to show how often a participant zaps a specific bot-player (indexed by color) compared to how often they experienced that bot-player's zap. This would assist the reader in understanding how these two different approaches to learning social norms would result in different behavior.

A4.

This is an important point, and I agree that a more detailed description of the computational models' performance and demonstrations of the differences between models' predictions and the way parameters values affect these predictions is needed. I added the graphs and demonstrations suggested by the reviewer to the supplementary materials. In addition, while preparing these materials it occurred to me that the name 'hybrid model' may not be accurate, and renamed the model 'Biased-Attribution' model.

The analysis below demonstrate how the generalization (or group-attribution) parameters G_Zap and G_Avoid set a limit on the level of generalization, which can contribute to asymmetry in learning about avoidance.

First, I plotted the average zapping rate for each of the four experimental conditions, each containing two blocks (benefit/harm action->omission and omission->action) in figure Rev6. As the trial-by-trial average was very noisy, for the sake of clearer visual presentation each participant's zapping timeline was smoothed with a moving window of 50 trials, before group level averaging. This allowed demonstration of the general trends of zapping and the way the models captured them. Model fitting was done with no such smoothing. On top of the participants' averaged zaps, I plotted the average (and smoothed) predictions made by all three models – individual, group, and hybrid (now renamed biased-attribution). It can be seen that all models captured the overall trends of zapping behaviour, which were mostly affected by the active/avoidance behaviour of the bot-players (i.e. changed on trial 70). It is also possible to observe the asymmetry in adaptation, as moving from harm-action to harm-omission led to the smallest change in zapping behaviour, compared with all other transitions, and that the biased-attribution model is better than other models in capturing it.

Figure Rev6: Zapping behaviour over time. The participants' trial-by-trial zapping behaviour was smoothed with a moving window of 50 trials, and averaged for each of the experimental conditions (Grey line). The models' trial-by-trial zap predictions were similarly smoothed and averaged, and are plotted on top of the participant's behaviour line. Note that while all models captured the main trend of participants' zaps, the biased-attribution model (red line) gave overall more accurate predictions, especially capturing the attenuated adaptation in the transition from harm-action to harm-omission norms (top left panel).

One reason for the overall similar performance of the three models is that zapping prediction is affected by the learned likelihood of other players to zap, where the models differ, and by the specific trial settings, the distance to stars and other players, which were the same for all models. To better understand the difference between the models I plotted the learned likelihood of other players to zap over time in Figure Rev7. The models were fitted to all 140 trials of each participant, and it is therefore possible to inspect the differences in learning about zaps and avoidances in all experimental conditions.

Figure Rev7 – Learning curves for zap-likelihood of the other players. The computational models differed in the way they learn and update the other players' likelihood of zapping according to their zaps and avoidances. The trial-by-trial average zap probabilities of all three bot-players were recovered after model-fitting, and averaged across participants. All three models show a similar pattern of increased and decreased likelihood according to the norms displayed by the bot-players, and show asymmetry in learning rates for zaps and avoidances. However, the biased-attribution model converged to higher zap likelihoods during omission norms than the individual and group models.

The plots show that all models estimated increased likelihood of zaps when the bot-players were following a zapping norm (harm-action and benefit-action), and reduced likelihood of zapping when the bot-players were following zap-avoidance norms. In addition, as all models included different learning rates for zaps and avoidances, all models demonstrate faster learning for zaps than for avoidances.

The models differ in the level to which they converge (the learning asymptote) during the zap-avoidance norm conditions. In these conditions, the group-learning converges to the lowest value most of the time, as complete group-level generalization facilitates learning. The biased-attribution model converged to higher likelihood values during the omission norms blocks and showed the most pronounced asymmetry in adaptation from harm-action to harm-omission conditions (top-left panel).

To better understand the effect that different model parameters' values have on learning about zap likelihood, I simulated the models with fixed values for the free parameters and the bot-players' zaps and avoidances experienced by participants in the four experimental conditions.

First, I examined the effect of changes to zap-avoidance learning rate, while zapping learning rate was fixed at 0.5. As demonstrated in Figure Rev8, when decreasing the value of the avoidance learning rate, the asymmetry in zap likelihood between the action norms and omission norms becomes more pronounced. It is important to note that as the action-norms conditions include both zaps and avoidances, the increased learning rates for avoidances affect learning in these conditions as well and leads to an overall reduction in estimation of zap likelihood. The simulations demonstrate the overall faster learning in group-level models, where the same number of learning opportunities affect the estimation of all players. In addition, the bias on generalization in the biased-attribution model set a limit to the level of zap and avoidance likelihood.

Figure Rev8 – Simulation of different zap-avoidance learning rate values. All three models were simulated with fixed zap learning rate of 0.5, and different zap avoidance learning rates. The simulation included the bot-players' zaps and avoidances from the two harmful zaps experimental conditions, moving from harm-omission to harm-action (top panels) and moving from harm-action to harm-omission (bottom panels). Note how changes to parameters contribute to asymmetry in adaptation between the top and lower panels.

In a separate step, I fixed the avoidance learning rate to 0.1, and changed the values of the zap learning rate (Figure Rev9). Here increased values of zap learning rates led to higher estimations of zap likelihoods in the action-norms conditions, which affected adaptation to omission-norm conditions. As the omission-norm conditions did not include zaps, changes to zap learning rate did not affect estimations when the omission-norm was the first norm experienced, as priors of the entire learning process were fixed across simulations.

Figure Rev9 - Simulation of different zap learning rate values. All three models were simulated with a fixed zap-avoidance learning rate of 0.1, and different zap learning rates. The simulation included the bot-players' zaps and avoidances from the two harmful zaps experimental conditions, moving from harm-omission to harm-action (top panels) and moving from harm-action to harm-omission (bottom panels). Note how changes to parameters contribute to asymmetry in adaptation between the top and lower panels.

Finally, to better understand the effect of the generalization parameters G_Zap and G_Avoid in the biased-attribution model, I changed one parameter's values while keeping the other parameter fixed at a value of 0.5 in a separate set of simulations (Figure Rev10). Changing the value of the generalization parameter changed the level to which zap likelihood converged. Generalization values of above 0.5 led to convergence to high zap likelihood, while values above 0.5 led to convergence to low zap likelihoods. Importantly, when these values were close to 1 for zap generalization and 0 for avoidance generalization, there was no bias in generalization. However, the model allows generalization to be biased and asymmetric, as were the results of the model fitting procedure presented in Figure 5 of the main manuscript. This can account for the lack of adaptation from harm-action to harm-omission norms, as negative zaps were readily generalized to all players and avoidance of negative zaps was not generalized. In the positive zap conditions, zap-avoidances were generalized more readily than active zaps.

Figure Rev10 - Simulation of different generalization parameters. In the top panels, G_{avoid} was fixed at 0.5 and different values were assigned to G_{zap} . In the bottom panels, G_{zap} was fixed at 0.5 and G_{avoid} changed across simulations. The simulation included the bot-players' zaps and avoidances from the two harmful zaps experimental conditions, moving from harm-omission to harm-action (left panels) and moving from harm-action to harm-omission (right panels). Note how changes to parameters contribute to asymmetry in adaptation between the right and left panels.

To better capture the effect of the generalization parameters on zap likelihood convergence I decided to rename the hybrid-model to biased-attribution model. While this model indeed includes both aspects of individual-level learning and group-level learning, the importance of the generalization mechanism in introducing a mechanism for asymmetry in learning (on top of the dual learning rates) may be better captured by the new model name.

Finally, the reviewer suggested demonstrating the relation between participants' zap rates of specific players and the how often they experienced that bot-player's zaps. I carried this analysis in response to the reviewer's next comment and the first reviewer comment related to first-hand and second-hand learning. I will therefore address it fully in the answer to comment 5.

5. In a related concern, I think that the reciprocity-based model would benefit greatly from the addition of a "direct" versus "indirect" reciprocity parameter. Extensive literature has demonstrated how direct (experiencing another bot zapping the participant) and

indirect (seeing the bot zap another bot) reciprocity account for different behavioral patterns (e.g., Rand & Nowak, 2013 for an overview). I believe this analysis would strengthen the results by examining whether there is a special emphasis in the learning process for observing a norm versus experiencing it.

A5:

Thank you for this comment, which was also raised by reviewer 1. The answers to both comments is therefore essentially the same.

Examining the different influences of first-hand experience and second-hand observation on learning is a very good suggestion. However, as I will show below, it may be a bit more complicated than simply expanding the computational models with the existing experimental design. The current experimental design, and indeed the main focus of the paper, have to do with the effect of behavioural prescription and experience on learning of social norms, while the specific group structure (who did what to whom) was not manipulated. This means that all bot-players had similar probabilities to zap others and zap the player. While it is possible to examine how the participants' zapping behaviour was related to first and second-hand experiences, it may not be very informative in the current design. I am currently running two follow-up experiments looking more closely at reciprocity and the role of first and second-hand information, which are based on the current experimental design and findings (see below).

To address the comment raised by the reviewer, I first expanded the hybrid learning model used in the manuscript to include different learning rates for experienced and observed zaps and avoidances:

First-hand update rule:

$$ZapProb_{t+1}^p = ZapProb_t^p + \begin{cases} LR_{Zap}^{Experience} \cdot (1 - ZapProb_t^p) & Z_t = 1 \\ LR_{Avoid}^{Experience} \cdot (0 - ZapProb_t^p) & Z_t = 0 \end{cases}$$

Second-hand update rule:

$$ZapProb_{t+1}^p = ZapProb_t^p + \begin{cases} LR_{Zap}^{Observe} \cdot (1 - ZapProb_t^p) & Z_t = 1 \\ LR_{Avoid}^{Observe} \cdot (0 - ZapProb_t^p) & Z_t = 0 \end{cases}$$

With the same learning rules used to generalize to other players, using the generalization parameters as in the hybrid model. I used a similar fitting procedure as the one used for the models in the main text. This model had a higher average DIC score in the negative zap conditions ($DIC_{New} - DIC_{Hybrid} = 6.6$, $t(129) = 14.28$, $p < 0.0001$), and was comparable with the hybrid model in the positive zap conditions ($DIC_{New} - DIC_{Hybrid} = -0.09$, $t(112) = -0.91$, $p = 0.36$). These results indicate that the addition of parameters aimed to account for first and second-hand learning did not increase the model fitting to the data sufficiently.

A description of the additional computational model and results is now included in the supplementary materials.

One possible cause for the reduced performance of this model may have to do with the current experimental design. Currently, participants experienced two sets of bot-players, one which did not zap at all and another that zapped from time to time. The models have to account for both sets of participants, and to the adaptation in behaviour when moving from one environment to another. If people behave differently within one environment than between environments, this model will be likely to underperform.

For example, in the current experiment participants were found to overall zap all bot-players with similar frequency, regardless of how much these players zapped overall in the harm-active conditions (green bars in figure Rev1). When moving to the new environment, where bot-players did not zap, zapping behaviour could not be associated with individual bot-player zaps, as there were no such zaps. This difference between conditions is accounted for by the models in this paper and is indeed the focus of this project.

However, when breaking the bot-players' zaps to first and second-hand zaps (yellow and blue bars respectively in the figure Rev1), a more complicated picture emerges. It seems that participants were more likely to zap players that zapped them the least, and most likely to zap players that zapped others the most. First and second-hand learning therefore affected behaviour differently in our task. Such within block differences are hard to account for in the current design.

Figure Rev1: Participants' zapping behaviour towards specific bot-player, according to their overall number of zaps (Green), number of zaps of the participant (Yellow), and number of zaps of other bot-players (Blue).

To demonstrate that unpacking this pattern may be beyond the scope of the current project, I added results from two follow-up experiments.

In figure Rev2, I present results from a follow-up experiment in which participants were playing with three bot-players displaying different zapping behaviours. Two bot-players were zappers, following the harm-action norm, and one was non-zapper, following a harm-avoidance pattern. It is clear from this figure that the fact that the harm-avoider did not zap even once was registered by the participants, and they tended to zap this player less than others. This can be seen in the first-hand zapping behaviour, showing the opposite pattern of what was observed when the player that zapped the participant the least (but at least once, in the current study) was zapped the most (in figure Rev1).

Figure Rev2: Results from follow-up experiment 1, where one bot-player(Min) follows a harm-avoidance pattern, while the two other players display a harm-action pattern.

A second follow-up study was designed to examine whether first-hand experiences are crucial for behavioural adaptation. In this study participants played with bot-players that displayed different behaviours toward each other, and toward the participant. They either zapped each other and avoided zapping the participant, or zapped the participant and avoided zapping each other. The results indicate that participants' behaviour was dependent on their first-hand experience – they zapped more when they were being zapped and avoided zapping when the players avoided zapping them (Figure Rev3, pink and purple bars).

Figure Rev3: Results from follow-up experiment 2, where the bot-players displayed different behaviour amongst themselves than toward the participant.

Taken together, these follow-up studies indicate that homogeneity of the norm behaviour, i.e. being displayed by most group members most of the time (Ullmann-Margalit, 2015), is important in the learning and adaptation of social norms.

The current work examined how behavioural features of social norms affect learning and adaptation, and therefore behaviours were displayed uniformly by the players. A more refined examination of the dependencies of social learning on the specific pattern of displayed behaviour (who do what to whom) could not be carried directly in the current settings. Using the same experimental framework, it is possible to examine more intricate social structure and dynamics, as was demonstrated in the follow-up studies.

A discussion on the limitation of current design in the study of more refined social learning strategies, and outline of future direction, was added to the discussion. The detailed description of the follow-up studies and the extra figures are not included in the revised manuscript or the supplementary materials.

Finally, I changed the name of the *Reciprocity* model to *Individual* model. This was done to highlight the main feature of the model, which is learning on individual level with no generalization to other players, but to avoid confusion with reciprocity as a first-hand experience.

Minor concerns:

1. Page 4, figure 1: it may be helpful to separate this figure based on the learning strategies (individual-specific or group generalization panels) to demonstrate the authors point more clearly

A1: I added labels to better distinguish between the strategies.

2. Page 6, lines 103-105: the prediction that “behaviors with aversive outcomes may be more readily generalized to all group members than helping behaviors” is not justified based on the literature presented.

A2: I rephrased this sentence (and added a relevant citation) to indicate that building on individual level attribution asymmetries, group level attributions may also be biased:

“For example, as negative moral behaviour is more readily attributed to an individual's character than positive behaviour (Mende-Siedlecki et al., 2013), behaviours with aversive outcomes may be more readily generalized to all group members than helping behaviours.”

3. Page 12, line 280: The prior refers to the probability of zapping before any experience, but the model includes two experimental blocks where the norm changes. Because participants presumably learn and update their prior after the first experimental block, it may be useful to specify two priors based on the order of norm sequences the participant received as another possible control for the sequential nature of the blocks.

A3: The prior indeed was a free parameter that was used to set the participants' expectation in the beginning of the first block. The prior for the second block was the average probability of zapping of all three players at the end of the first block, i.e. their experience in the first block changed their expectations for the next block. This was the case for the group-level and biased-attribution models. In the individual-level model the priors for the second block were the same as in the first block, as the main assumption was that there was no generalization in expectation from one player to another. This clarification is now included in the description of the models.

The effect of the priors in the different experimental groups, and the changes in expectation from the beginning of the first block to the beginning of the second block, can be observed in figure Rev7, showing the learning patterns of each model. The learning in the new block starts off where the learning in the previous block ended. Figure Rev7 is now included in the supplementary materials.

As the reviewer suggested, I also included a summary of the estimated variables according to the different block order (Table Rev3 below). These demonstrate that there in most cases the order of blocks did not lead to big changes in parameter estimation (for example for learning rates). However, the priors were different between the different orders, with higher prior zap probability estimated for the harm-omission->harm-action condition than for the harm-action->harm omission condition. Interpretation of these effects is not straightforward, and again examining the learning curves in figure Rev7 can help understanding how these priors affected estimation later in the task. Table Rev3 is now included in the supplementary materials.

	w_0	LR_{Zap}	LR_{Avoid}	w_1	w_2	w_3	G_{Zap}	G_{Avoid}	$Prior$
Harm-Omission -> Harm-Action	-4.74 +0.08	0.42 +0.01	0.11 +0.01	1.82 +0.16	-4.47 +0.21	4.90 +0.20	0.62 +0.02	0.61 +0.01	0.51 +0.02
Harm-Action -> Harm-Omission	-4.30 +0.07	0.41 +0.01	0.14 +0.01	1.26 +0.16	-2.28 +0.19	4.29 +0.24	0.61 +0.01	0.53 +0.01	0.40 +0.01
Benefit-Omission -> Benefit-Action	-5.27 +0.12	0.42 +0.01	0.31 +0.01	1.59 +0.18	-1.70 +0.24	4.53 +0.26	0.52 +0.01	0.69 +0.01	0.46 +0.02
Benefit-Action -> Benefit-Omission	-4.79 +0.10	0.47 +0.01	0.30 +0.02	1.04 +0.23	-1.94 +0.23	2.94 +0.28	0.46 +0.01	0.72 +0.01	0.53 +0.01

Table Rev3 – Parameter estimation for the four experimental conditions.

4. Page 14, line 315: the author may want to exclude participants who never zapped in their behavioral results as well if they are going to be removed from the modeling, because the modeling results inform the behavioral.

A4: I think that it is important to keep the participants that did not zap at all in the initial behavioural analysis, as not zapping at all is a valid behaviour in the task. Indeed, zap avoidance was a marker of adaptation in the omission norms. The main reason for removing these participants from the model fitting procedure was to avoid unreliable parameter estimation when fitted behaviour had no variability. This is now stated in the text. Importantly, the number of participants that never zapped was relatively small (28, which are 10% of participants), and such participants were excluded from all conditions.

As suggested, I conducted the mixed-effects ANOVA on the subset of participants that zapped more than once, and were used for modeling (248 out of 276). The results were essentially the same:

An ANOVA was used to analyse the individual adaptation patterns, with Zap-Behaviour Order (Action First/Omission First), Zap-Outcome (Benefit /Harm) and their interaction as main effects. A significant Zap-Behaviour Order effect was found ($F(1,244) = 13.01$, $p = 0.0004$, $partial \eta^2 = 0.05$), indicating that participants displayed higher levels of adaptation when moving from an omission norm to an action norm. In addition, a significant interaction was found between Zap-Norm Order and Zap-Outcome ($F(1,244) = 6.16$, $p = 0.014$, $partial \eta^2 = 0.024$), with no significant Zap-Outcome effect ($F(1,244) = 0.02$, $p = 0.88$, $partial \eta^2 < 0.0001$). These results are now included in the supplementary materials.

5. Page 14, line 319: “simple reciprocity learning rule” seems like a misnomer as the reciprocity rule is more complex than the group rule since it requires updating three separate players whereas the group rule only needs to update a single group value. At minimum the reciprocity learning rule requires more working memory and is more complex is that respect.

A5: I agree, and changed this statement.

6. Page 17, line 390: I think the author should avoid this claim since the resistant norm being the active behavior and harmful outcome for others is likely context specific.

A 6: This statement is now refined to reflect that this is the combination of active behaviour and harmful outcome was found to contribute uniquely in the social norms examined in this study. However, I further suggest that this combination can be a factor in making real-life social norms persistent, and should be considered among other mechanisms, for example the way such norms are imposed and maintained by social institutions (formal and informal), social signaling role of following norms and habits.

7. Page 18, lines 431-452: The claims about economic games only taking a “snapshot of participant’s tendencies at one point” is unfounded. Economic games are often used for both learning paradigms and dynamics of repeated play. The claims that economic games have “limited set of behaviors and norms ... focusing on ... cooperating or defecting” is unnecessary and simply untrue. There are numerous studies employing economic games to examine behaviors relating to trust, generosity, punishment, etc. The claim that this was “a social setting which participants have less experience, a video game, rather than monetary transaction tasks that are familiar” is unfounded. In general, the paragraph makes several unsubstantiated claims and framing this paragraph about the positives of the author’s paradigm may be better suited.

A7: I agree with the reviewer and completely revised this paragraph, to stress the positive aspects of the paradigm, as well as its limitations.

8. Page 19, line 458: I’m not sure what this sentence means: “a mixture of individual and group-level learning was shown to make some norms more resilient than others.” The author should clarify where this conclusion arises.

A8: This sentence was meant to indicate the asymmetry in group-level attribution. I changed it to read: Another mechanism was a bias in group-level attribution, where behaviours with negative outcomes to others were more readily attributed to other group members than behaviours with positive outcomes.

Tracked Changes version of the paper

Learning how to behave: Cognitive learning processes
account for asymmetries in adaptation to social norms

Short title:

Cognitive Learning of Social Norms

Uri Hertz

Department of Cognitive Sciences, University of Haifa, Haifa, Israel

Correspondence:

Uri Hertz

Orcid: 0000-0003-4852-3516

Department of Cognitive Sciences,

University of Haifa

Haifa, Israel, 3498838

Phone: (+972) 4 6146239

Email: uhertz@cog.haifa.ac.il

Keywords: Social Norms, Reinforcement Learning, Social Cognition

Abstract

Changes to social settings caused by migration, cultural change or pandemics force us to adapt to new social norms. Social norms provide groups of individuals with behavioural prescriptions, and therefore can be inferred by observing their behaviour. This work aims to examine how cognitive learning processes affect adaptation and learning of new social norms. Using a multiplayer game, I found that participants initially complied with various social norms exhibited by the behaviour of bot-players. After gaining experience with one norm, adaptation to a new norm was observed in all cases but one, where an active-harm norm was more resistant to adaptation. Using computational learning models, I found that active behaviours were learned faster than acts of omission, and that harmful behaviours were more readily attributed to all group members than learned on a group level, while beneficial behaviours were learned on an individual level. These results provide a cognitive foundation for learning and adaptation to descriptive norms and can inform future investigations of group-level learning and cross-cultural differences and social adaptation.

Introduction

Social norms are the unwritten rules that prescribe and guide behaviour within a society and with which group members generally comply [1–3]. Social norms govern a group's behaviour, are manifested in the behaviour of most individuals most of the time, and may change between social groups and over time. For example, the norm governing how we greet each other when we meet can differ quite arbitrarily from one culture to the next, or during global events such as the COVID-19 pandemic (Figure 1). Adhering to group norms can ensure cooperation within a group [2,4], make social conduct more predictable [5] and signal one's group affiliation to others [6]. Failure to learn and adapt might unintentionally send the wrong signals through inappropriate behaviour that may lead to frustration, isolation, resentment and intergroup distress [7]. While the challenge of learning and adapting to new social norms has been studied from the perspective of the social structures and mechanisms supporting socialization [8] as well as from an evolutionary, normative point of view [1,2], far less attention has been devoted to the contribution of social cognitive learning mechanisms to this problem.

Figure 1 – Learning a new social norm's behavioural prescription

When introduced to a new social setting, one may need to adapt one's behaviour according to the prevalent social norm. (A) Such social norms stochastically govern the behaviour of individuals in a group, affecting most group members most of the time. Newcomers (e.g., the person in blue plaid in the figure) can infer the social norm through accumulated observations and experiencing of interactions between group members, and can adapt their behaviour accordingly. (B) Such learning can occur at a group level (top right), i.e. generalizing attributing the behaviour to all

group members across individuals, or on an individual level (bottom-right), i.e. learning only about specific individuals.

Formatted: Font: 10 pt, Complex Script Font: 10 pt, Highlight

Social norms change how individuals behave. Unlike such norms include injunctive norms, which indicate how people *should* behave, and descriptive norms indicate how other people behave [9], which are at the focus of this work. Descriptive norms have been shown to affect people's behaviour, for example when exposed to other people's recycling habits [10], finance management [11] or alcohol use [12], and are the focus of this work. Such norm effects have also been observed in lab experiments, notably in the seminal works on social influence and conformity by Sherif [13] and Asch [14] regarding perceptual decisions. Other studies have shown that people adapt their behaviour and preferences after learning about others' preferences [15–17], indicating the importance of social information in forming one's own behaviour and beliefs [18–20]. While it is possible to explicitly state a descriptive norm, in many cases people form their perception of norms on their own [21]. The effects of social norms on behaviour may therefore rely on how people learn about others' behaviour and form such a descriptive norm.

Formatted: Font: Italic, Complex Script Font: Italic, Highlight

One way to learn about a descriptive social norm is by observing the behaviour of members of a group, and accumulating such observations over time [22–26] (Figure 1A). Such accounts borrow from non-social computational models of associative and reinforcement learning [22]. For example, when learning about a person's honesty, one may observe whether a person gives truthful advice over time, increasing the estimation of her honesty when she gives accurate advice, and decreasing it when she gives misleading advice [27]. When learning about groups, learners may use the same learning mechanisms, learning about specific individuals in a group, and adjust their behaviour according to the specific partner they encounter. However, learners may learn a group-level trait, attributing observations from individuals to all group members, indicating learning about a social norm that governs the group's behaviour [28–31] (Figure 1B).

In the literature concerning learning about action-outcome associations, such as Pavlovian and operant conditioning, the strength of associative learning is often mapped to two dimensions – the appetitive/aversive outcome of an action, and the active/passive

nature of the action [32,33]. For example, one may learn to increase a pattern of behaviour after it has been actively rewarded, or when it leads to the omission of an aversive response (avoiding punishment). Similarly, the omission of an appetitive outcome and receiving punishment may lead a learner to reduce the likelihood of displaying a behaviour pattern. While these contingencies may rely on similar computational principles, they are known to be processed differently. Punishments and rewards are processed by different neural mechanisms [34], and can have different effects on learning. ~~In a similar manner~~ Similarly, omission and action are perceived and processed differently [35,36]. Such asymmetries can therefore give rise to different biases in learning, and shape the way people learn and adapt to social norms. This work seeks to examine how features of the behavioural prescription of social norms affect adaptation to these norms and the learning process constraints underlying such effects. Specifically, it is hypothesised that due to constraints of the cognitive learning mechanisms, behavioural features of social norms will make some norms easier to attain and harder to relinquish in favour of new norms. One constraint has to do with the perceptual aspects of learning, as some behaviours are more readily detected than others, ~~for example, e.g.~~ action vs omission. In addition, the transfer from individual-level learning to group-level learning may be influenced by the norm's behavioural prescriptions ~~For example, as negative moral behaviour is more readily attributed to an individual's character than positive behaviour~~ [37] ~~for example~~ behaviours with aversive outcomes may be more readily generalized to all group members than helping behaviours.

To study these hypotheses, I adapted the appetitive/aversive and action/omission dimensions to the domain of social norms, using norms that prescribe behaviour that can benefit/harm others through action/omission acts (Figure 2). In a sequential social dilemma paradigm called the star-harvest game [38], participants collected stars and could sacrifice a move to zap other players. In different experimental conditions, zap outcomes were either harmful or beneficial to other players. The participants were exposed to different social norms displayed by the behaviour of three bot-players. The action/omission dimension was formed by the bot-players' active zapping or zap avoidance behaviour (Figure 2). Different combinations of these features formed different types of norms, which were characterized by different behavioural prescriptions. The Harm-Action norm was marked by active zaps that had negative outcomes for

others, i.e. zapping a player who is on your route to a star. The Harm-Omission norm was manifested in avoidance of negative zaps. The Benefit-Action norm was manifested in active zaps that had positive outcomes for others, while avoidance of positive zaps was a manifestation of the Benefit-Omission norm. This allowed examination of how participants learn and adapt to social norms and which social norms persist when moving to a new social environment.

Methods

Participants

The Amazon M-Turk platform was used to recruit 276 participants for this study, including 157 men (age: mean \pm std: 35.45 \pm 9.75) and 119 women (age: 39.12 \pm 11.1). Participants were randomly assigned to one of four experimental conditions, which differed in the order of experimental blocks and **in** the zap outcome (positive/negative). I aimed for at least 60 participants in each of the four conditions based on estimation of **an** effect size of 0.5 **for a within-participants difference in zapping rates between Harm-Omission and Harm-Action conditions, based on a pilot of 20 participants (not included in this study)**. Due to the random assignment and to ensure there were enough participants in each condition, the number of participants in each block differed slightly (Harm-Action First N = 76, Harm-Omission First: N = 71, Benefit-Omission First: N = 63, Benefit-Action First: N = 66). No participant was excluded from the analyses. All participants provided informed consent and received monetary compensation at a fixed rate of 3.5 USD for 15 minutes of participation. The study was approved by the research ethics committee at the Faculty of Social Sciences at the University of Haifa, Israel (number 038/18).

Star-Harvest Game

The star-harvest game was developed to provide a flexible and rich setting in which multiple types of social norms can be displayed in a user-friendly manner. The game included four players, represented by coloured squares that move around a 10x10 grid (Figure 2, see example here: <http://socialdecisionlab.net/resources.html>). The game is played on a turn-by-turn basis, and the order of players remains constant throughout the game. In each turn the players could either move in one of four directions and collect stars that appeared on the grid, or zap by emitting a pink ray in one of the four directions. Players caught in the ray in the negative zap outcome conditions were sent to

a 'time-out zone' visible to the player for three turns. Those caught in the ray in the positive zap outcome conditions received a small bonus star. After each round in which all players took a turn, a new star could appear somewhere on the grid with a 0.75 probability, and uncollected stars could disappear. Each player's collected stars appeared in their 'score' section on the screen. **The participants did not receive any bonus based on the stars they collected beyond the fixed monetary rate.**

Social Norms Algorithms

The behaviour of the bot-players was governed by algorithms implementing different social norms. In each experimental condition, the behaviour of all three bot-players was governed by the same algorithm. A short description of the different algorithms is given below, and a detailed description of the algorithms is provided in supplementary materials.

All bot-players began each turn by looking for stars. If they were the player closest to a star, they would move towards it. Otherwise, when the zap outcome was negative,

Harm-Action bot-players zapped other players that were on their way to a star, while

Harm-Omission bot-players would move away **from other players without zapping.**

When the zap outcome was positive, **Benefit-Omission** bot-players would also move away without zapping anyone. **Benefit-Action** bot-players would start every turn with a probability of zapping others, even if they were closest to a star. **This probability was**

dependent on their distance from the closest star, and decreased the closer they were to the star (distance of 1 was associated with a zap probability of 0.02).

Analysis

Statistical analyses were carried using Matlab R2018b (Mathworks Inc., USA). The Markov Chain Monte Carlo (MCMC) Metropolis-Hastings algorithm was used for model fitting and estimation for each participant [39]. For model comparisons, for each model a Deviance Information Criterion (DIC) [40] was calculated for each individual. I used in-house Matlab code and an MCMC toolbox for Matlab developed by Marko Laine [41].

Results

Participants played the star-harvest game online, where they moved across a 2D grid using arrows and collected stars that appeared (and disappeared) from time to time, with three other bot-players (Figure 2). Participants were randomly assigned to one of the

four experimental conditions. Each experimental condition began with one experimental block in which the three bot-players displayed one of the four norms (Harm-Action, Harm-Omission, Benefit-Action, Benefit-Omission), followed by a second block with a new set of bot-players, marked by changes in the players' colours, which displayed a different norm. Each experimental block included 70 turns **for each player**. To make the experimental instructions consistent, the zap's outcome did not change between blocks, such that the norms displayed by the bot-players changed from Harm-Action to Harm-Omission (and vice versa) or from Benefit-Action to Benefit-Omission (and vice versa).

Figure 2 – Experimental design – The Star-Harvest Game

Participants played the star-harvest game online. (A) The game layout consisted of four players who moved across a 2D grid and collected stars, with players marked by coloured squares. The participant in this case played the blue square (marked P), playing against three other bot-players. In each trial, players could either move using the blue arrows or zap using the arrows and the Zap button. (B) Players could either zap each other by sending a ray that affected other players or avoid zapping other players. (C) The zap outcome was either harmful, sending the zapped player to a time-out zone for three turns, or beneficial, such that the affected player received a small star worth a tenth of a regular star. (D) The algorithms governing the behaviour of the bot-players led to four different social norms with different behavioural prescriptions. (E) Participants played two experimental blocks, with different bot-players displaying different norms. The zap outcome remained consistent over the two blocks.

The main behavioural marker of adaptation to social norms was the percentage of times participants zapped other players when they had the opportunity to do so, i.e., when they shared a column or row with another player (see numbers of zaps and zap opportunities in supplementary materials). Adaptation to the norm would result in lower zapping rates when the bot-players avoid zapping (Harm-Omission and Benefit-Omission norms) than when bot-players actively zap others (Harm-Action and Benefit-Action norms). In the first experimental block, the participants in all conditions adapted their zapping behaviour to the behaviour of their surroundings (Figure 3). This adaptation was quantified using an ANOVA, with zapping percentages as the dependent variable and Zap Outcome (Harm/Benefit) and Zap Behaviour (Action/Omission) and their interactions as main effects. I found a significant effect of Zap Behaviour ($F(1, 272) = 35.92$, $p < 0.0001$, $partial \eta^2 = 0.115$), indicating that participants were more likely to zap others when they were in the company of other zappers. In addition, participants were more likely to zap others when zaps were associated with harmful outcomes ($F(1,272) = 37.09$, $p < 0.0001$, $partial \eta^2 = 0.122$), indicating a bias towards competitive behaviour in such video-game scenarios. The interaction between Zap Behaviour and Zap Outcome was not significant ($F(1,272) = 2.29$, $p = 0.13$, $partial \eta^2 = 0.008$). These results indicate that in the first experimental block, lacking prior experience in the task, participants generally learned and adapted to all social norms.

The next behavioural analysis examined adaptation to new social norms between the first and the second experimental blocks by subtracting each participant's zapping rate in the omission norm block from the action norm block. When this measure was positive, it indicated high behavioural adaptation between conditions, in line with the change in norms. When it was close to zero, it indicated low behavioural adaptation. An ANOVA was used to analyse the individual adaptation patterns, with Zap-Behaviour Order (Action First/Omission First), Zap-Outcome (Benefit /Harm) and their interaction as main effects. A significant Zap-Behaviour Order effect was found ($F(1,272) = 12.65$, $p = 0.0004$, $partial \eta^2 = 0.044$), indicating that participants displayed higher levels of adaptation when moving from an omission norm to an action norm. In addition, a significant interaction was found between Zap-Norm Order and Zap-Outcome ($F(1,272) = 5.73$, $p = 0.017$, $partial \eta^2 = 0.02$), with no significant Zap-Outcome effect ($F(1,272) = 0.00002$, $p = 0.97$, $partial \eta^2 < 0.0001$). As can be seen in the averaged zap-rate graph

(Figure 3), participants showed adaptations in all conditions but one—when moving from a Harm-Action norm to a Harm-Omission norm, giving rise to the interaction effect.

Figure 3 – Adaptation to new social norms

Participants played the star-harvest game under four experimental conditions, in two blocks that differed according to the social norms displayed by the bot-players. In each experimental condition, the percentage of times participants zapped others when they had the opportunity to do so was examined. In the first experimental block, participants adapted their zapping behaviour and matched the zapping norm around them. In the second block, participants adapted their behaviour to the new norm during all transitions except the transition from Harm-Action norm to Harm-Omission norm (red to yellow bars). On all the graphs, grey dots represent individual scores, bars represent the mean, and error bars represent the standard error of the mean.

The next analysis steps were aimed at examining potential learning mechanisms that underlie the behavioural adaptation patterns, using computational learning models. The models were used to examine how zap behaviour (action/omission) and zap outcome (benefit/harm) are treated by a learner, in line with the asymmetries in adaptations observed so far. In addition, the models were designed to examine whether and how observation of one player's behaviour are used to infer group-level norms (Figure 4). Specifically, the models included **reciprocity individual-level-based** learning, **occurring only in individual-level, social group-level-norm** learning, **where learning is done only in group-level**, and a hybrid **biased-attribution** model that allows weighted level of **transfer attribution** from individual to group level (Supplementary materials and Figure 4).

All models were aimed at predicting the participants' decision to zap a target player, i.e., a player that shares a column or row with the participant. This decision on each trial t was logistically dependent on a number of variables (Eq. 1): The participant's overall tendency to zap, his current distance from a star (variable $StarDist$), his current distance from the target player (variable $TargetDist$), and the estimated zapping behaviour of this target player (the probability that the target would zap other players, parameter $ZapProb$). The contribution of these variables to the decisions was determined by a set of free parameters $\{w_0, w_1, w_2, w_3\}$, and the value of these variables was calculated in each turn. These weights were used to model the cost associated with zapping, as they allow the availability of stars to overcome the tendency to zap others.

$$[1] \quad p_t(\text{Zap a target}) \sim w_0 + w_1 \cdot StarDist_t + w_2 \cdot TargetDist_t + w_3 \cdot ZapProb_t^{Target}$$

The distance to stars and targets can be calculated directly from the data available in each turn. However, the target player's zapping behaviour, i.e., the probability that the target player would zap other players, had to be learned from observations and interactions in previous turns. This learning mechanism differed between models (Figure 4). In all models, no learning was done if the observed player did not have an opportunity to zap anyone, i.e., did not share a row or a column with any player, or if the observed player was the closest player to a star and moved toward it. In addition, the models were fitted to the data with no information regarding the outcome of the zaps, harmful or beneficial, and were affected only by the estimated likelihood of zapping and distance to stars and targets.

The first model ~~was assumed that a reciprocity model, where~~ learning occurred only at the individual-level. When observing player p 's zap (or avoidance), the learner updates his belief about the likelihood of player p to zap in the future (Figure 4). When player p zaps another player at time t , the variable Z_t is set to 1, and when player p avoids zapping (he had the opportunity but did not zap), Z_t is set to 0. A prediction error is calculated between Z_t and the previous estimation of the player's zapping probability, $ZapProb_t^p$, and is used to update this probability with different learning rates for zap and avoidance. Zapping probabilities for all players were initially set by another free parameter $Prior$. To account for asymmetry in learning, the model included different learning rates for action (zaps) and omission (avoidance).

$$[2] \quad ZapProb_{t+1}^p = ZapProb_t^p + \begin{cases} LR_{Zap} \cdot (1 - ZapProb_t^p) & Z_t = 1 \\ LR_{Avoid} \cdot (0 - ZapProb_t^p) & Z_t = 0 \end{cases}$$

The second type of model assumed a complete attribution to group-level, was a social norm learning model, where each observation is used to update a group level zap probability which applies to all players, $ZapProb_t^{Group}$. Such transfer can speed up learning and adaptation to new norms, as it accumulates information across all players, and is especially useful when displays of the new norm's behaviour are sparse [29].

$$[3] \quad ZapProb_{t+1}^{Group} = ZapProb_t^{Group} + \begin{cases} LR_{Zap} \cdot (1 - ZapProb_t^{Group}) & Z_t = 1 \\ LR_{Avoid} \cdot (0 - ZapProb_t^{Group}) & Z_t = 0 \end{cases}$$

Figure 4 – Models of learning about others' behaviour

After observing a player's behaviour, the participants may update their beliefs about other players' likelihood to zap, which would inform their future decisions whether to zap others. Three learning models were suggested. In the reciprocity-individual model, learning is done on the individual-level, and the behaviour of the observed player is used to update belief about his zap behaviour with value Z_t , with learning rate LR_{Zap}/LR_{Avoid} depending on the observed behaviour (zap/avoidance). In the norm-group model, one player's behaviour is used to update all players' zap probability with the same values and learning rates. In the hybrid biased-attribution model,

the observed player's and the other players' zap behaviour are both updated with the same learning rates but with different values.

The third model was a hybrid **biased-attribution** model that included the **reciprocity individual** learning mechanism (equation 2) and an additional group-level updating to all other players. This is captured by two free parameters $\{G_{Zap}, G_{Avoid}\}$, which specify how much each observed zap or avoidance behaviour of one player **contributes-is attributed to the group, i.e.** to the updating of the zap probability of the other players. When G_{Zap} is set to 1, it increases the zap probability of other players as if these players were doing the zapping, converging with the **norm-group-learning** model. When G_{Avoid} is set to 1, it decreases the zap probability of other players as if they avoided zapping, again converging with the norm learning model. However, when $\{G_{Zap}, G_{Avoid}\}$ are close to 0.5 the transfer is non-informative **essentially setting an expectation that other players are just as likely to zap or avoid. Asymmetries in the generalization parameters would lead to biased attribution to group level, as is demonstrated in a set of simulations in the supplementary materials.**

$$[4] ZapProb_{t+1}^{others} = ZapProb_t^{others} + \begin{cases} LR_{Zap} \cdot (G_{Zap} - ZapProb_t^{others}) & Z_t = 1 \\ LR_{Avoid} \cdot ((1 - G_{Avoid}) - ZapProb_t^{others}) & Z_t = 0 \end{cases}$$

All models were fitted to each participant's decisions (zap/avoid) across both experimental blocks (see methods and Supplementary materials). **To avoid unreliable parameter estimation,** only participants who zapped at least once were included in this analysis (N = 248 of 276). **In addition, the mixed-effect ANOVA analysis of adaptation in zap behaviour was carried on the same subset of participants with no changes in the results from the main analysis see supplementary materials.** In a series of model comparisons, taking into account both model fit to the data and the number of parameters used by the model, the **hybrid-biased-attribution** model was found to significantly outperform other models (see Table S1 **including a model with different parameters for direct and indirect reciprocity [42]**). This result indicates that our participants did not use a **simple** reciprocity learning rule, but were flexible in the way they update beliefs about other players, i.e., group level or norm inference, from observation of single players.

The model fitting procedure allowed estimation of all free parameters for all participants, facilitating overall evaluation of these parameters and comparing them between groups of participants (Table 1). The weights assigned to each factor affecting zapping (w_0, w_1, w_2, w_3) all significantly differed from 0 in both the positive and negative zap-outcome conditions (Table 1). Overall, participants were averse to zaps (negative w_0), more so when zapping had a positive outcome than when it had a negative outcome ($p = 0.04$, see Table 1). Participants were more likely to zap when stars were far away (positive w_1), indicating the cost of zaps. They were more likely to zap targets that were close to them (negative w_2), more so when zaps had harmful outcomes than when they had beneficial outcomes ($p = 0.01$, Table 1). These results indicate that the participants were sensitive to the task settings in each turn, their distance to stars and to other players, and these affected their decision to zap other players.

Figure 5 – Learning parameters underlying asymmetric adaptation to social norms

The **hybrid biased-attribution learning** model that best fit participants' decisions to zap included two mechanisms for learning from observing other players' zaps or zap avoidance. (A) The model included different learning rates for action (zap) and omission (zap avoidance) behaviours.

Learning rates were lower for omission than for action, indicating an asymmetry in the contribution of active behaviour to norm learning. (B) The model included different group-level **attribution transfer** parameters for omission (zap avoidance) and action (zaps). Group-level **transfer-attribution values** were higher for behaviours that had an aversive contingency – Harm-Action and Benefit-Omission norms. In all the graphs, grey dots represent individual scores, bars represent the mean, and error bars **show** the standard error of the mean.

The **hybrid-biased-attribution** model also indicated that participants were affected by other players' likelihood of zapping (positive w_3). This value was learned by observing the players' behaviour over time. The model included two learning rates, for action (zap) and for omission (zap avoidance) behaviours (Figure 4A). The effects of Zap Behaviour (action/omission, **within-within**-subjects), Zap Outcome (harm/benefit, between subjects) and their interaction on learning rates, were examined using a mixed-effects ANOVA, with individually estimated learning rates (LR_{Zap}, LR_{Avoid}) as independent variables. A significant Zap Behaviour effect was observed ($F(1,529) = 79.81, p < 0.0001, partial \eta^2 = 0.23$), such that learning rates for action (zaps) were higher than for omission (avoidance). In addition, a significant Zap Outcome effect was found ($F(1,529) = 19.96, p < 0.0001, partial \eta^2 = 0.07$), such that participants in the beneficial zap conditions had higher overall learning rates. Finally, a significant interaction effect was observed ($F(1,529) = 14.83, p = 0.0001, partial \eta^2 = 0.052$), such that learning from Harm-Omission behaviour was associated with lower learning rates than learning from Benefit-Omission behaviour.

In addition, two parameters were estimated for group-level **transfer-attribution** of information from the observed player to all other players (Figure 5B). The effects of Zap Behaviour (action/omission, within subjects), Zap Outcome (harm/benefit, between subjects) and their interaction on **transfer-group-level attribution**, were examined using a mixed-effects ANOVA, with the individually estimated group-level transfer parameters as independent variables. A significant effect of Zap Behaviour was observed, such that omission (avoidance) behaviours were associated with higher group-level transfer values ($F(1,529) = 10.53, p = 0.0013, partial \eta^2 = 0.038$). Zap Outcome did not have a significant effect ($F(1,529) = 0.1, p = 0.75, partial \eta^2 < 0.001$). A significant interaction was observed ($F(1,529) = 28.98, p < 0.0001, partial \eta^2 = 0.099$), pointing to higher group-level **transfer-attribution** of behaviours with aversive contingencies: Harm-Action and Benefit-Omission.

Negative Zap Conditions (N = 136)									
	w_0	LR_{Zap}	LR_{Avoid}	w_1	w_2	w_3	G_{Zap}	G_{Avoid}	Prior
Estimate	-4.5 ± 0.078	0.41 ± 0.012	0.12 ± 0.01	1.53 ± 0.15	-3.36 ± 0.21	4.59 ± 0.22	0.62 ± 0.015	0.57 ± 0.014	0.45 ± 0.014

Mean ± SEM									
t(df = 135)	-31.8			5.42	-8.98	11.39			
p	<0.0001			<0.0001	<0.0001	<0.0001			
Positive Zap Conditions (N = 113)									
	w_0	LR_{zap}	LR_{Avoid}	w_1	w_2	w_3	G_{zap}	G_{Avoid}	$Prior$
Estimate									
Mean ± SEM	-5 ± 0.11	0.44 ± 0.013	0.3 ± 0.015	1.3 ± 0.21	-1.8 ± 0.23	3.74 ± 0.27	0.49 ± 0.014	0.71 ± 0.013	0.49 ± 0.015
t(df = 112)	-23.36			3.23	-3.87	6.88			
p	<0.0001			0.0016	0.0002	<0.0001			

Table 1 - Parameter estimations of the hybrid learning model separately for the negative and positive outcome conditions.

Discussion

The aim of this study was to investigate how cognitive learning mechanisms account for learning and adaption to new social norms. Specifically, I examined how two features of the behavioural prescription of norms, its manifestation in action or omission and the outcome of this behaviour, whether beneficial or harmful, affect learning and adaptation. Using a multiplayer star-harvest game in which the behaviour of three bot-players was governed by algorithms that implemented four different social norms, I examined how people learn new social norms and how their experience with one norm affects adaptation in the transition from one norm to another. I found that on their first encounter with the task, participants learned and adapted to the social norm displayed by the bot-players. Yet, in the second block of the experiment, when a new social norm was displayed by a new set of bot-players, their previous experience affected their adaptation. Specifically, the norm manifested in Harm-Action behaviour persisted when participants faced a new set of Harm-Omission players, while in all other transitions, participants adapted their behaviour. The resistant norm was characterized both by an active behaviour and by a harmful outcome for others, implying a competitive intent. This combination seems to contribute to the unique persistence of **this social norms in the current experimental design.**

Computational modelling of social learning proposed a mechanistic explanation for the observed behaviour, and indicated that social learning in the task went beyond reciprocal individual-level learning. The best-best fitted model, the hybrid-biased attribution model, suggested that participants' decisions to zap or avoid zapping other players were influenced by several parameters, amongst them the distance from stars, indicating the cost of zapping, and the estimation of the target player's likelihood to zap. This estimation was based on the specific player's previous behaviour, in line with the individual-level learning mechanism, and also incorporated other players' previous behaviour, in line with group-level social-norm inference. The weight given to other players' behaviour, or the magnitude of group-level transferattribution, was a free parameter in the model. Behaviours that carried an aversive contingency, omission of benefit or harmful action, were found to be more readily transferred-attributed to group-level-other-players. Such group-level transfer facilitates learning as it allows rapid accumulation of sparse behaviours across players, instead of accumulating them independently for each participant. Behaviours with beneficial contingency were associated with low group-level generalizationattribution, suggesting more personal and reciprocity-based learning for prosocial behaviours [43]. In addition, learning rates associated with actions were consistently higher than for omissions, in line with non-social learning findings, further supporting the observed asymmetry in adaptation [36,44]. Both mechanisms can work together to make behavioural prescriptions persistent even when social settings change, attenuating adaptation to new social norms.

The results of this study are in line with previous findings on social learning of individuals' traits and behaviour, and demonstrate how these are linked to group-level inference findings. On the individual level, research has shown that people are quick to infer about bad social behaviour from sparse data, as such negative behaviours are deemed more diagnostic of a person's moral character [37,45]. In addition, actions were shown to be more readily attributed and indicative of a person's general character than acts of omission, as they are both more likely to be detected and less likely to be explained away (plausible deniability) [36]. Social learning of individuals' traits was shown to be important to form predictions about others' behaviour and to adapt one's behaviour accordingly [24,46]. Beyond inferring from one's behaviour in a specific situation about his general trait, people also infer from one person to all other group members [30,47]. Adults and children can attribute a set of behaviours to all other group

members, mostly when such group membership is salient [31,48]. The current results indicate that social learning about others' behaviour can be set on a continuum, with some behaviours more readily attributed than others on the individual level (action vs. omission), and some more readily generalized to indicate group-level norm (harmful vs. beneficial contingencies). A unified cognitive learning framework can account for both types of social learning, operating simultaneously for individual and group-level inference, and affecting adaptation and one's future behaviour.

The current study examines adaptation to norms in new, unfamiliar surroundings, the star harvest game, and the effect of experience on adaptation to new norms. It ~~therefore~~, ~~therefore~~, examines dynamic, quick, behavioural adaptation. It is a departure from studies aimed at characterising cooperation and prosocial behaviour as a stable trait [49–51] or from examining gradual changes across development and acculturation [7,52]. ~~The current study's approach is limited in the sense that the learned social norms may not represent a long-lasting behaviour or tendency, as it does not rely on real-life contexts, such as monetary or resource sharing, which are common in the study of social norms [28,53]. However, the current paradigm allows control of the effect of experience on social adaptation, and a rich and flexible lab model of social learning. As such, it may be useful for understanding cross-cultural differences in adaptation to social norms and the contribution of cognitive learning processes and cultural background (previous experience) to this process [53–55].~~

~~Some limitations arise from the use of bot-players instead of live-interaction with humans. Participants were not given explicit information regarding the identity of the other players, either if these were humans or bots. As the experiments were taking place online, where it is possible to interact anonymously both with other humans and with algorithmic bots, supported this ambiguity. While it is possible that during interactions with humans the patterns observed here will be amplified or different, participants' behaviour in the positive-zaps conditions, where zaps were mainly to benefit others and had no clear benefit for the participant, suggests that participants did treat the other players as if they were fellow participants, to some extent. Another limitation of the current experimental design was that the bot-player's' behaviour was homogenous, following the notion that social norms are carried by most group members, most of the times [2]. This meant that it was hard to distinguish between learning from first-hand experience (direct reciprocity) and ~~from~~ second-hand observations (third-party~~

reciprocity) [42], and examining how people learn in a non-homogenous environment with different people displaying different norms. However, future studies may build on the current paradigm and manipulate the rate of first and second-hand experiences, and the homogeneity of behaviour, as well as introducing groups and coalitions, to examine different social learning dynamics and their interaction with cognitive learning mechanisms [56,57].

Such studies usually rely on pre-existing dispositions and use behavioural tools such as economic games to take a snapshot of participants' tendencies at one point, relying on people's experience with monetary transactions. While the effect of experience on behaviour in such games was demonstrated in a spill-over effect, as people who experienced a positive and cooperative environment tended to cooperate in other contexts [56], there are some limitations to the use of monetary games to study social norms. First, they have a limited set of behaviours and norms, with most games focusing on two strategies only, cooperating or defecting. In addition, there is high variability in how people from different cultures play these games, which are tightly related to the different social norms prevailing in their culture [48,51,55].

To conclude, this study aimed to provide a cognitive-learning perspective on the problem of learning and adaptation to social norms. The behavioural results indicated asymmetries in the learning of social norms, and the computational models indicated two mechanisms that may underlie these asymmetries. One such mechanism was an omission bias in learning, whereby actions were more readily learned than omissions. Another mechanism was addition, a bias in group-level attribution, where behaviours with negative outcomes to others were more readily attributed to other group members than behaviours with positive outcomes. A mixture of individual and group-level learning was shown to make some norms more resilient than others. These mechanisms may influence adaptation to social norms outside the lab, making social norms whose behavioural manifestations are active and harmful more persist even when social settings change. Finally, the experimental approach used here can be elaborated to account for many different norms, and the use of principles and computational frameworks from cognitive learning can inform future investigations of cross-cultural differences and adaptation to descriptive social norms.

Data Availability

Formatted: Font: (Default) Arial, Font color: Custom Color(RGB(34,34,34)), Complex Script Font: Arial, English (United States), Pattern: Clear (White), Highlight

All datasets and code used to analyse the data are available at: <https://osf.io/f6erm/>

A demo of the star-harvest game is available at:

<http://socialdecisionlab.net/resources.html>

Acknowledgment

The author thanks Prof. Chirs D. Frith for his comments and feedback on an earlier version of this manuscript. UH was supported by the Israel Science Foundation (1532/20).

References

1. Fehr E, Schurtenberger I. 2018 Normative foundations of human cooperation. *Nat. Hum. Behav.* **2**, 458–468. (doi:10.1038/s41562-018-0385-5)
2. Ullmann-Margalit E. 2015 *The emergence of norms*. Oxford University Press, USA.
3. Elster J. 1989 *The cement of society: A survey of social order*. Cambridge university press.
4. Fehr E, Williams T. 2017 Creating an Efficient Culture of Cooperation. *Work. Pap.* **7041**, 1–35.
5. FeldmanHall O, Shenhav A. 2019 Resolving uncertainty in a social world. *Nat. Hum. Behav.* **3**, 426–435. (doi:10.1038/s41562-019-0590-x)
6. Brewer MB. 1991 The Social Self: On Being the Same and Different at the Same Time. *Personal. Soc. Psychol. Bull.* **17**, 475–482. (doi:10.1177/0146167291175001)
7. Berry JW. 1997 Immigration, acculturation, and adaptation. *Appl. Psychol.* **46**, 5–34.
8. Macionis JJ, Benoit C, Jansson M. 2000 *Society: the basics*. Prentice Hall Upper Saddle River, NJ.
9. Bicchieri C, Muldoon R, Sontuoso A. 2011 Social norms.
10. White KM, Smith JR, Terry DJ, Greenslade JH, McKimmie BM. 2009 Social influence in the theory of planned behaviour: The role of descriptive, injunctive,

- and in-group norms. *Br. J. Soc. Psychol.* **48**, 135–158.
(doi:10.1348/014466608X295207)
11. Kast F, Meier S, Pomeranz D. 2018 Saving more in groups: Field experimental evidence from Chile. *J. Dev. Econ.* **133**, 275–294.
(doi:10.1016/j.jdeveco.2018.01.006)
 12. Borsari B, Carey KB. 2003 Descriptive and injunctive norms in college drinking: a meta-analytic integration. *J. Stud. Alcohol* **64**, 331–341.
(doi:10.15288/jsa.2003.64.331)
 13. Sherif M. 1935 A study of some social factors in perception. *Arch. Psychol. (Columbia Univ.)* **27**, 1–60.
 14. Asch SE, Guetzkow H. 1951 Effects of group pressure upon the modification and distortion of judgments. *Groups, leadership, men*, 222–236.
 15. Julian W, Hackel LM, Van Bavel JJ. 2018 Shifting Prosocial Intuitions: Neurocognitive Evidence for a Value Based Account of Group-based Cooperation. *PsyArXiv* (doi:10.31234/osf.io/u736d)
 16. Campbell-Meiklejohn DK, Bach DR, Roepstorff A, Dolan RJ, Frith CD. 2010 How the opinion of others affects our valuation of objects. *Curr. Biol.* **20**, 1165–70.
(doi:10.1016/j.cub.2010.04.055)
 17. Garvert MM, Moutoussis M, Kurth-Nelson Z, Behrens TEJ, Dolan RJ. 2015 Learning-Induced Plasticity in Medial Prefrontal Cortex Predicts Preference Malleability. *Neuron* **85**, 418–428. (doi:10.1016/j.neuron.2014.12.033)
 18. Festinger L. 1954 A Theory of Social Comparison Processes. *Hum. Relations* **7**, 117–140. (doi:10.1177/001872675400700202)
 19. Bandura A, Walters RH. 1977 *Social learning theory*. New York: Prentice-hall Englewood Cliffs, NJ.
 20. Csibra G, Gergely G. 2006 Social learning and social cognition : The case for pedagogy. *Process. Chang. Brain Cogn. Development*, 249–274.
(doi:10.1.1.103.4994)
 21. Larimer ME, Neighbors C. 2003 Normative misperception and the impact of

- descriptive and injunctive norms on college student gambling. *Psychol. Addict. Behav.* **17**, 235–243. (doi:10.1037/0893-164X.17.3.235)
22. Lockwood PL, Klein-Flügge MC. 2020 Computational modelling of social cognition and behaviour—a reinforcement learning primer. *Soc. Cogn. Affect. Neurosci.* , 1–11. (doi:10.1093/scan/nsaa040)
 23. Farmer H, Hertz U, Hamilton AF de C. 2019 The neural basis of shared preference learning. *Soc. Cogn. Affect. Neurosci.* **14**, 1061–1072. (doi:10.1093/scan/nsz076)
 24. Tamir DI, Thornton MA. 2018 Modeling the Predictive Social Mind. *Trends Cogn. Sci.* **xx**, 201–212. (doi:10.1016/j.tics.2017.12.005)
 25. FeldmanHall O, Dunsmoor JE. 2019 Viewing Adaptive Social Choice Through the Lens of Associative Learning. *Perspect. Psychol. Sci.* **14**, 175–196. (doi:10.1177/1745691618792261)
 26. Hackel LM, Mende-Siedlecki P, Amodio DM. 2020 Reinforcement learning in social interaction: The distinguishing role of trait inference. *J. Exp. Soc. Psychol.* **88**, 103948. (doi:10.1016/j.jesp.2019.103948)
 27. Bellucci G, Molter F, Park SQ. 2019 Neural representations of honesty predict future trust behavior. *Nat. Commun.* **10**, 5184. (doi:10.1038/s41467-019-13261-8)
 28. Hackel LM, Wills JA, Van Bavel JJ. 2020 Shifting prosocial intuitions: Neurocognitive evidence for a value-based account of group-based cooperation. *Soc. Cogn. Affect. Neurosci.* , 1–15. (doi:10.1093/scan/nsaa055)
 29. Kleiman-Weiner M, Saxe R, Tenenbaum JB. 2017 Learning a commonsense moral theory. *Cognition* (doi:10.1016/j.cognition.2017.03.005)
 30. Hamilton DL, Sherman SJ. 1996 Perceiving persons and groups. *Psychol. Rev.* **103**, 336–355. (doi:10.1037/0033-295X.103.2.336)
 31. Rhodes M. 2012 Naïve Theories of Social Groups. *Child Dev.* **83**, 1900–1916. (doi:10.1111/j.1467-8624.2012.01835.x)
 32. Huys QJM, Cools R, Gölzer M, Friedel E, Heinz A, Dolan RJ, Dayan P. 2011 Disentangling the Roles of Approach, Activation and Valence in Instrumental and

- Pavlovian Responding. *PLoS Comput. Biol.* **7**, e1002028. (doi:10.1371/journal.pcbi.1002028)
33. Heyes CM. 1994 Social Learning in Animals: Categories and Mechanisms. *Biol. Rev.* **69**, 207–231. (doi:10.1111/j.1469-185X.1994.tb01506.x)
 34. Palminteri S, Pessiglione M. 2017 Opponent brain systems for reward and punishment learning: causal evidence from drug and lesion studies in humans. In *Decision Neuroscience*, pp. 291–303. Elsevier.
 35. Hunt LT, Rutledge RB, Malalasekera WMN, Kennerley SW, Dolan RJ. 2016 Approach-Induced Biases in Human Information Sampling. *PLoS Biol.* **14**, 1–23. (doi:10.1371/journal.pbio.2000638)
 36. Ritov I, Baron J. 1995 Outcome Knowledge, Regret, and Omission Bias. *Organ. Behav. Hum. Decis. Process.* **64**, 119–127. (doi:10.1006/obhd.1995.1094)
 37. Mende-Siedlecki P, Baron SG, Todorov A. 2013 Diagnostic Value Underlies Asymmetric Updating of Impressions in the Morality and Ability Domains. *J. Neurosci.* **33**, 19406–19415. (doi:10.1523/JNEUROSCI.2334-13.2013)
 38. Leibo JZ, Zambaldi V, Lanctot M, Marecki J, Graepel T. 2017 Multi-agent Reinforcement Learning in Sequential Social Dilemmas. *Proc. 16th Conf. Auton. Agents MultiAgent Syst.*, 464–473.
 39. Kruschke J. 2015 *Doing Bayesian data analysis: A tutorial introduction with R JAGS, and Stan*. 2nd edn. Elsevier.
 40. Spiegelhalter DJ, Best NG, Carlin BP, Van Der Linde A. 2002 Bayesian measures of model complexity and fit. *J. R. Stat. Soc. Ser. B (Statistical Methodol.* **64**, 583–639. (doi:10.1111/1467-9868.00353)
 41. Haario H, Laine M, Mira A, Saksman E. 2006 DRAM: Efficient adaptive MCMC. *Stat. Comput.* **16**, 339–354. (doi:10.1007/s11222-006-9438-0)
 42. Rand DG, Nowak MA. 2013 Human cooperation. *Trends Cogn. Sci.* **17**, 413–425. (doi:10.1016/j.tics.2013.06.003)
 43. Mahmoodi A, Bahrami B, Mehring C. 2018 Reciprocity of social influence. *Nat. Commun.* **9**, 2474. (doi:10.1038/s41467-018-04925-y)

44. Tsetsos K, Chater N, Usher M. 2012 Saliency driven value integration explains decision biases and preference reversal. *Proc. Natl. Acad. Sci. U. S. A.* **109**, 9659–64. (doi:10.1073/pnas.1119569109)
45. Martijn C, Spears R, Van Der Pligt J, Jakobs E. 1992 Negativity and positivity effects in person perception and inference: Ability versus morality. *Eur. J. Soc. Psychol.* **22**, 453–463. (doi:10.1002/ejsp.2420220504)
46. Koster-Hale J, Saxe R. 2013 Theory of Mind: A Neural Prediction Problem. *Neuron* **79**, 836–848. (doi:10.1016/j.neuron.2013.08.020)
47. Levy SR, Stroessner SJ, Dweck CS. 1998 Stereotype formation and endorsement: The role of implicit theories. *J. Pers. Soc. Psychol.* **74**, 1421–1436. (doi:10.1037/0022-3514.74.6.1421)
48. Kalish CW. 2012 Generalizing norms and preferences within social categories and individuals. *Dev. Psychol.* **48**, 1133–1143. (doi:10.1037/a0026344)
49. Crockett MJ, Kurth-Nelson Z, Siegel JZ, Dayan P, Dolan RJ. 2014 Harm to others outweighs harm to self in moral decision making. *Proc. Natl. Acad. Sci.* **111**, 17320–17325. (doi:10.1073/pnas.1408988111)
50. Rand DG, Greene JD, Nowak M a. 2012 Spontaneous giving and calculated greed. *Nature* **489**, 427–430. (doi:10.1038/nature11467)
51. Cohn A, Maréchal MA, Tannenbaum D, Zünd CL. 2019 Civic honesty around the globe. *Science (80-.)*. **365**, 70 LP – 73. (doi:10.1126/science.aau8712)
52. Jordan JJ, McAuliffe K, Warneken F. 2014 Development of in-group favoritism in children's third-party punishment of selfishness. *Proc. Natl. Acad. Sci.* **111**, 12710–12715. (doi:10.1073/pnas.1402280111)
53. Bowles S *et al.* 2008 "Economic man" in cross-cultural perspective: Behavioral experiments in 15 small-scale societies. *Behav. Brain Sci.* **28**, 795–855. (doi:10.1017/s0140525x05000142)
54. Heyes CM, Frith CD. 2014 The cultural evolution of mind reading. *Science (80-.)*. **344**, 1243091–1243091. (doi:10.1126/science.1243091)
55. Henrich J, Heine SJ, Norenzayan A. 2010 Beyond WEIRD: Towards a broad-

based behavioral science. *Behav. Brain Sci.* **33**, 111–135.
(doi:10.1017/S0140525X10000725)

56. FeldmanHall O, Son J-Y, Heffner J. 2018 Norms and the Flexibility of Moral Action. *Personal. Neurosci.* **1**. (doi:10.1017/pen.2018.13)
57. Gershman SJ, Pouncy HT, Gweon H. 2017 Learning the Structure of Social Influence. *Cogn. Sci.* **41**, 545–575. (doi:10.1111/cogs.12480)